# The gut microbiome in konzo

Matthew S. Bramble[1,15], Neerja Vashist[1,2,15], Arthur Ko [3], Sambhawa Priya[4], Céleste Musasa [1], Alban Mathieu[5], D' Andre Spencer[1], Michel Lupamba Kasendue[6], Patrick Mamona Dilufwasayo [1,6], Kevin Karume [1,6], Joanna Nsibu[6], Hans Manya [1,6], Mary N. A. Uy [1,7], Brian Colwell [8], Michael Boivin [9], J. P. Banae Mayambu[10], Daniel Okitundu[11], Arnaud Droit [5], Dieudonné Mumba Ngoyi [6,12], Ran Blekhman[4], Desire Tshala-Katumbay [6,13✉] & Eric Vilain[1,2,14✉]

Konzo, a distinct upper motor neuron disease associated with a cyanogenic diet and chronic malnutrition, predominately affects children and women of childbearing age in sub-Saharan Africa. While the exact biological mechanisms that cause this disease have largely remained elusive, host-genetics and environmental components such as the gut microbiome have been implicated. Using a large study population of 180 individuals from the Democratic Republic of the Congo, where konzo is most frequent, we investigate how the structure of the gut microbiome varied across geographical contexts, as well as provide the first insight into the gut flora of children affected with this debilitating disease using shotgun metagenomic sequencing. Our findings indicate that the gut microbiome structure is highly variable depending on region of sampling, but most interestingly, we identify unique enrichments of bacterial species and functional pathways that potentially modulate the susceptibility of konzo in prone regions of the Congo.

[1] Center for Genetic Medicine Research, Children's Research Institute, Children's National Hospital, Washington, DC, USA. [2] Department of Genomics and Precision Medicine, George Washington University School of Medicine and Health Sciences, Washington, DC, USA. [3] Department of Medicine, David Geffen School of Medicine, University of California, Los Angeles, Los Angeles, CA, USA. [4] Departments of Genetics, Cell Biology, and Development, University of Minnesota, Minneapolis, MN, USA. [5] Computational Biology Laboratory, CHU de Québec - Université Laval Research Center, Québec City, QC, Canada. [6] Institut National de Recherche Biomédicale (INRB), Kinshasa, DR, Congo. [7] College of Medicine, University of the Philippines, Manila, Manila, Philippines. [8] School of Public Health, Texas A&M University, College Station, TX, USA. [9] Department of Psychiatry and Neurology & Ophthalmology, Michigan State University, East Lansing, MI, USA. [10] Ministry of Health National Program on Nutrition (PRONANUT), Kinshasa, DR, Congo. [11] Centre Neuro-Psychopathologique (CNPP), University of Kinshasa, Kinshasa, Congo. [12] Department of Tropical Medicine, University of Kinshasa, Kinshasa, DR, Congo. [13] Department of Neurology and School of Public Health, Oregon Health & Science University, Portland, OR, USA. [14] International Research Laboratory of Epigenetics, Data, Politics, Centre National de la Recherche Scientifique, Washington, DC, USA. [15]These authors contributed equally: Matthew S. Bramble, Neerja Vashist. ✉email: tshalad@ohsu.edu; evilain@CNMC.org

O ur current understanding of the symbiotic relationship between humans and the gut microbiome, is largely based on findings from western industrialized nations. Few studies to date have investigated the structure and potential role of the gut flora in African and other non-western societies[1–5]. Collectively, studies on the microbiome have furthered our understanding of basic bacterial composition and relationships that are associated with geographic setting[6,7], host genetics[8–10], age[11,12], nutrition[13,14], disease[15–18], and to a large extent, dietary practices[4,19–21]. Abundance of certain bacterial genera has been demonstrated to be associated with different lifestyle practices and geographical locations. Urbanized populations for example are enriched for *Bacteroides* and conversely, *Prevotella* species are more common in the guts of humans in rural subsistence living environments[4,5,7,22,23]. While the exact factors behind these distinctions are uncertain, long-term diet, food diversity, and overall nutrition are likely to be important contributors.

The Democratic Republic of the Congo (DRC) is one of the least developed countries in the world with a high percentage of individuals relying on a monotonous cassava (*Manihot esculenta Crantz*) diet for basic survival. Cassava, also known as yucca or manioc, is a drought-tolerant plant which resists harsh environmental conditions including poor and arid soils in tropical regions. As such, it is an important crop for subsistence and source of calories for populations dwellings in these regions. Consumption of improperly processed food derived from bitter cassava, which harbors high levels of cyanogenic compounds such as linamarin, can result in an irreversible nonprogressive motor neuron disease known as konzo, that predominately manifests in children and women of childbearing age[24,25]. While certain risk factors, such as food insecurity, chronic malnutrition, and particularly a lack of sulfur containing amino acids are associated with outbreaks of konzo, the exact biological mechanisms underlying disease susceptibility and severity remain poorly understood[25]. The consumption of toxic plants for survival is not uncommon in other mammalian species such as the Giant Panda and Desert Wood Rat, who consume foods laced with high levels of cyanogenic glucosides and toxic creosote, respectively. However, these mammals have evolved a gut microbiome composition that serves to aid in the detoxification of these xenobiotics, effectively enabling the survival of such species[26–28]. Few populations in the world rely exclusively on toxic foods for survival, making the DRC a unique country to query the influence of a detrimental subsistence on the gut flora and its relationship to this debilitating multifactorial neurological disease.

Here we present a large comparison of gut microbiome profiles in children from the Democratic Republic of the Congo, using shotgun metagenomic sequencing, with study populations ranging from the urbanized capital of Kinshasa to the extremely rural settings of south-western DRC, including children affected with konzo from prone villages. These data expand on our understanding of the gut microbiome in non-western lifestyles, as well as serve as the first investigation into the gut microbiome of populations that rely on toxic cassava as their staple food source. Additionally, these data reveal an enrichment of bacteria and genes in the konzo prone regions of the DRC that may exacerbate the effects of cyanogenic glucosides by enhancing linamarase activity, the key enzyme needed for the hydrolysis and subsequent release of cyanide in the human gut.

## Results

**Study population description.** During March of 2018, we collected fecal samples and dietary recall questionnaires from 180 individuals in the Democratic Republic of the Congo; 30 from

Kinshasa, 30 from a rural village of Masi-Manimba and 120 from konzo prone regions in Kahemba (Fig. 1). Samples from both the populous urban capital of Kinshasa (Kin) and Masi-Manimba (Mas), which is ~300 km east of the capital, were taken from presumably healthy children who were not affected with konzo. It should be noted that while outbreaks of konzo have not been documented in Masi-Manimba, residents of this region of the DRC also have a very high reliance on cyanogenic cassava as a staple food source. The Kahemba region, which is ~600 km South East of Kinshasa, harbors villages with the varying degrees of konzo outbreak frequency, as well as being the region with the most cases of konzo in the country. Our research team surveyed 2 villages in the Kahemba Health Zone that have historically had higher prevalence of konzo (HPZ) cases, as well as a village with lower prevalence of the disease (LPZ). Samples and dietary questionnaires were collected from 30 unaffected children from the HPZ (UHPZ) as well as 30 konzo-affected children from the HPZ (KHPZ), in addition to 30 unaffected children from the LPZ (ULPZ) and 30 konzo-affected children from the same village (KLPZ) (Supplementary Data 1). Individuals with konzo were diagnosed by in-country medical experts familiar with this disease and the signatures associated with such, following the WHO criterion for diagnosis. While unaffected children in Kahemba did not have konzo at the time of collection, they were chronically under-nourished and should not be thought of as "healthy" per se, as their susceptibility to konzo remains a possibility. Dietary questionnaires highlighted that food diversity was highest in the urban capital and very low in the Kahemba region, where protein sources such as meat and dairy products were generally not consumed in the week prior to specimen collection (Supplementary Fig. 1). These findings were in line with previous reports unveiling monotonous protein-deficient cassava diets in the region of Kahemba[29,30].

**Overall gut microbiome characteristics.** After filtering to include bacterial taxonomic assignments that were present at greater than or equal to 0.01% relative abundance in each individual, we observe that all study groups regardless of living environments harbored on average over 450 unique bacterial species (Fig. 2a) (Supplementary Data 2). All study groups also displayed measures of α-diversity as measured by the Shannon index that were indictive of a diverse microbial ecosystem (Fig. 2b). While variable between groups, the four most abundant bacterial phyla, as expected for human populations, were *Bacteroidetes*, *Firmicutes*, *Proteobacteria*, and *Actinobacteria* (Supplementary Fig. 2a) (Supplementary Data 2). Additionally, *Bacteroidia* and *Clostridia* were the two most abundant classes in all groups (Supplementary Fig. 2b) (Supplementary Data 2) with *Bacteroidales* and *Clostridiales* being the dominating bacterial orders in all study populations (Supplementary Fig. 2c) (Supplementary Data 2). When assessing bacteria at the family taxonomic rank, we saw more broad differences between study populations. The gut flora of individuals from urban center of Kinshasa are dominated by bacteria belonging to the family *Bacteroidaceae* (Kin: 20.2%). The predominate bacterial family for rural populations residing in Masi-Manimba, and the high konzo prevalence zone of Kahemba is *Prevotellaceae* (Mas: 18.5%, UHPZ: 20.8%, KHPZ: 20.5%) (Supplementary Fig. 2d) (Supplementary Data 2). However, groups living in the low konzo prevalence zone of Kahemba regardless of disease status are dominated by *Lachnospiraceae* (ULPZ: 14.7%, KLPZ: 15.4%) (Supplementary Fig. 2d) (Supplementary Data 2). At the genus level, we also observe trends associated with urban or rural living environments. The study participants from Kinshasa harbor *Bacteroides* (Kin: 21.6%) as the most abundant genus, while the genus *Prevotella* is the most abundant for those residing in the rural settings of Masi-Manimba

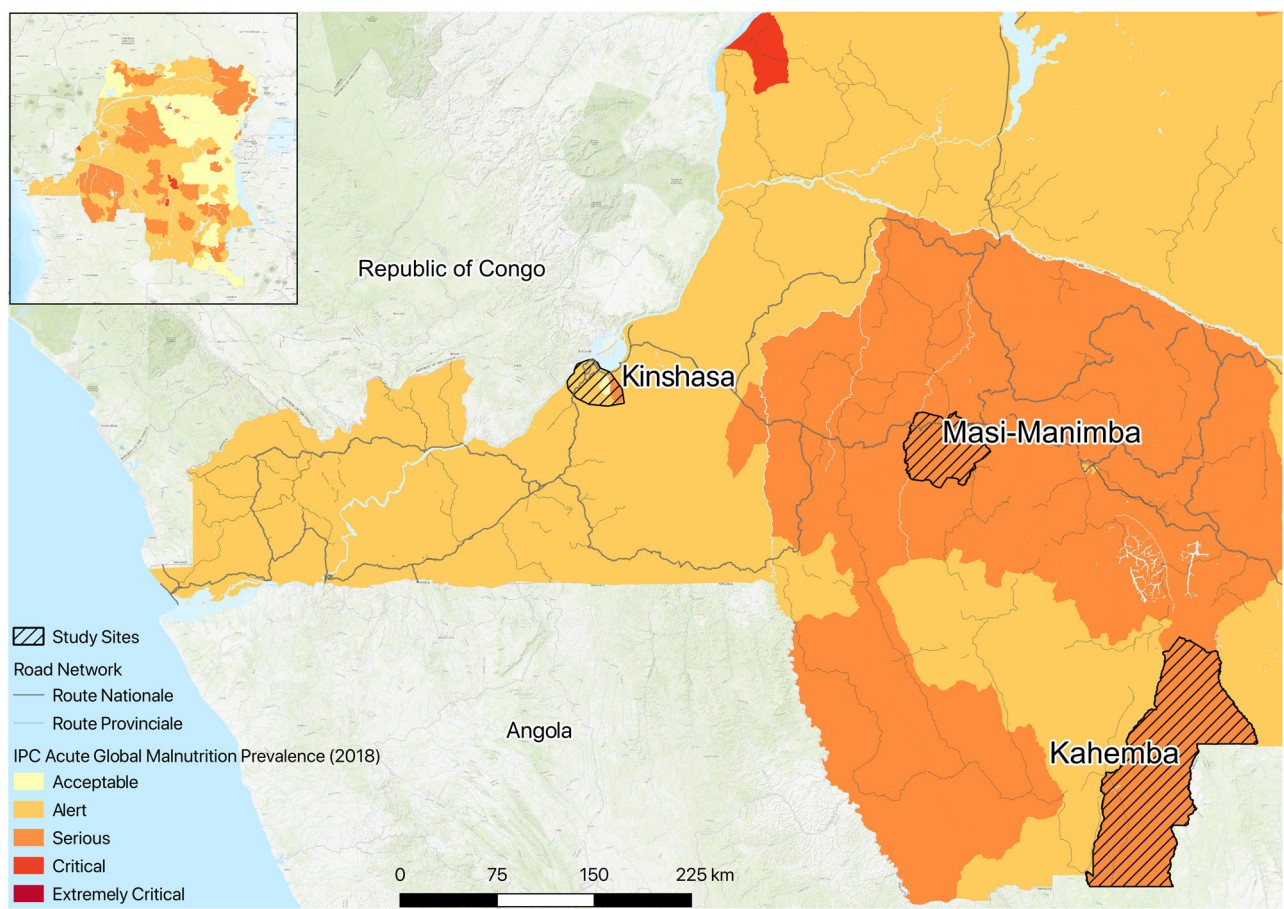

**Fig. 1 Map of DR Congo highlighting sampling locations and food insecurity.** Sampling locations and summary of study populations from South West DRC that includes the urban capital of Kinshasa ($n = 30$, age $= 8.7 \pm 1.66$, 15 F, 15 M), rural regions in Masi-Manimba ($n = 30$, age $= 9.9 \pm 2.32$, 15 F, 15 M), and 2 Konzo prone villages in Kahemba (Unaffected Low Prevalence Zone (ULPZ), $n = 30$, age $= 7.93 \pm 2.32$, 15 F, 15 M) (Konzo Low Prevalence Zone (KLPZ), $n = 30$, age $= 8.33 \pm 2.67$, 12 F, 18 M) (Unaffected High Prevalence Zone (UHPZ), $n = 30$, age $= 9.03 \pm 2.03$, 15 F, 15 M) (Konzo High Prevalence Zone (KHPZ), $n = 30$, age $= 9.63 \pm 2.31$, 12 F, 18 M). Using qGIS 3.8 software, we generated the map illustrating the current status of food insecurity for children 6–59 months old in the DRC at the health zone level. Data and shapefiles were extracted from available datasets from Humanitarian Data Exchange, which is coordinated through OCHA. Using the most recent and available administrative boundary data as a geographic base, we overlaid the August 2018 to June 2019 Integrated Food Security Phase Classification (IPC) data provided by the OCHA DR-Congo. This dataset represents the estimated prevalence of Global Acute Malnutrition (GAM), the weight to height ratio, of children 6–59 months in the representative health zones.

and the Kahemba HPZ (Mas: 19.8%, UHPZ: 22.7%, KHPZ: 22.6%) (Fig. 2c) (Supplementary Data 2). Unaffected adolescents from the Kahemba LPZ also are dominated by the genus *Prevotella* (ULPZ: 16.4%); however, those with konzo from the same zone have a roughly equal relative abundance of *Bacteroides* (KLPZ: 15.6%) and *Prevotella* (KLPZ: 15.5%) (Fig. 2c) (Supplementary Data 2). Despite varied abundances, all individuals residing outside of urban Kinshasa on average have a significantly higher Prevotella to Bacteroides ratio (Supplementary Fig. 2e). After filtering to include bacterial species that ≥0.01% average relative abundance in any of the six groups, we were left with 694 species of interest, of which show distinct abundance profiles and cluster based on geographic/village location (Fig. 2d) (Supplementary Data 2). Given the high level of individual gut microbiome variability that has been documented, we also assessed intra-group bacterial dissimilarity using the Bray-Curtis index and observed that children from Kinshasa are collectively the most variable as a group, whereas those individuals residing in the rural Kahemba HPZ regardless of disease status are the most similar overall to one another (Supplementary Fig. 2f). These data suggest that factors contributing to the gut microbiome profiles are likely more uniform in rural regions as compared to urban settings of the DRC.

**Gut microbiome profiles and functional potential segregate across geographic locations**. After accounting for all possible interactors such as age, sex, location, and disease status, our data indicate that geographic location (cassava toxicity) is the variable that significantly contributes to observed bacterial composition differences. When assessing geographic location differences (intergroup differences) which most importantly coincide with dietary practices, we observe that the gut microbiome profiles of those in Kinshasa compared to unaffected children from all rural locations are significantly different based on Bray-Curtis dissimilarity measures. When comparing the gut microbiome abundance profiles of the Kinshasa group to the unaffected children of the rural regions of Masi-Manimba and Kahemba, we see that these bacterial profiles significantly segregate at the genus taxonomic rank (PERMANOVA $p = 1 \times 10^{-5}$) (Fig. 3a). For this global urban versus rural comparison, the abundance of the genus *Prevotella* is most strongly associated with the first principal coordinate (Axis.1) values (Spearman $\rho = 0.68$, $p = 1 \times 10^{-12}$), which accounts for 30.7% variability, while the abundance of *Faecalibacterium* is most associated with second principal coordinate values (Axis.2) (Spearman $\rho = -0.75$, $p = 1 \times 10^{-12}$) accounting for 18.9% of overall variability (Fig. 3a). In more specific comparisons between Kinshasa vs.

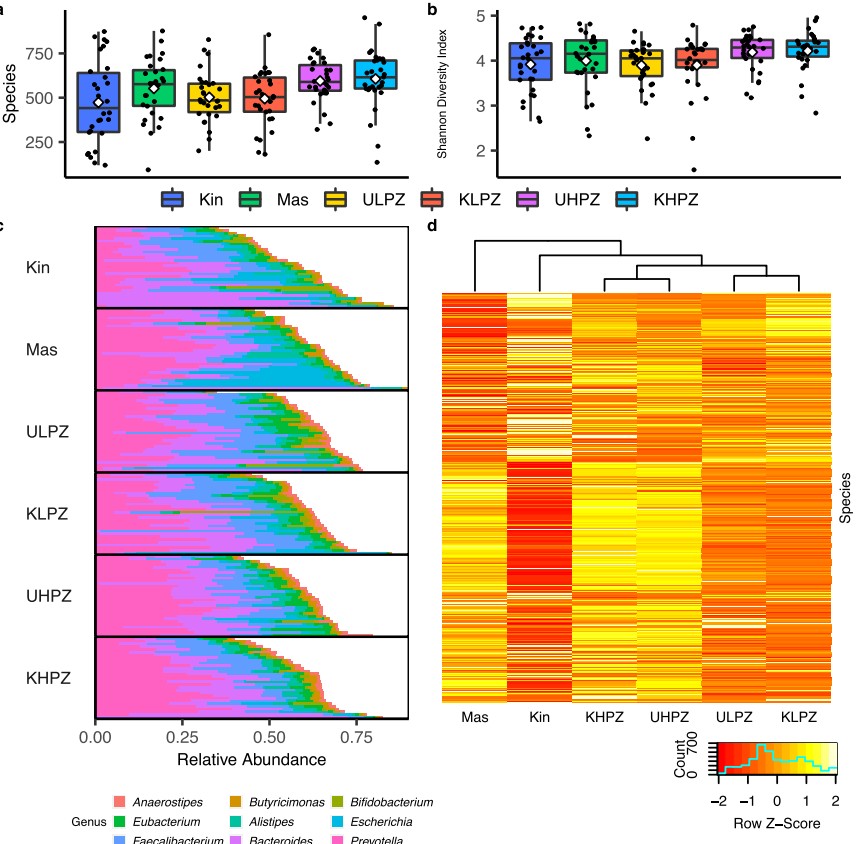

**Fig. 2 Overall alpha diversity and bacterial distribution in study groups.** Microbiome composition for all study groups that include **a** species level assignments post filtering to include those bacteria whose relative abundance ≥0.01% in each of the 180 participants from Kinshasa (Kin) ($n = 30$, mean = 473.2), Masi-Manimba (Mas) ($n = 30$, mean = 552.1), Unaffected Low Prevalence Zone (ULPZ) ($n = 30$, mean = 502.4), Konzo Low Prevalence Zone (KLPZ) ($n = 30$, mean = 494.3), Unaffected High Prevalence Zone (UHPZ) ($n = 30$, mean = 594.5), and Konzo High Prevalence Zone (KHPZ) ($n = 30$, mean = 606.2). **b** Shannon Index measures post filtering that includes species in each participant that had a relative abundance ≥0.01 from Kinshasa ($n = 30$, mean = 3.918), Masi-Manimba ($n = 30$, mean = 3.996), ULPZ ($n = 30$, mean = 3.897), KLPZ ($n = 30$, mean = 3.9), UHPZ ($n = 30$, mean = 4.186), and KHPZ ($n = 30$, mean = 4.217). **c** Highly abundant genus level assignments in the study groups (standard deviation for genus measures can be found in Supplementary File 2). **d** Z-score Heat map representation of the average relative abundances of the 694 species that passed the ≥0.01% relative abundance in either of the six study groups. In **a** and **b**, data are represented as boxplots where the diamond denotes the mean, middle line in the box is the median, the lower hinge is the first quartile, the upper hinge is the third quartile, and the whiskers extend from the lower and upper hinges to the smallest and largest value, respectively, at most to 1.5 * IQR (IQR, interquartile range, is the distance between the first and third quartile), with each individual value plotted.

Masi-Manimba (Fig. 3b), Kinshasa vs. unaffected children in the LPZ (Fig. 3c) or Kinshasa vs. unaffected children in the HPZ (Fig. 3d), strong segregation remains, with varying degrees of statistical significance (PERMANOVA $p = 2 \times 10^{-5}$, $p = 0.00139$, $p = 1 \times 10^{-5}$, respectively). Interestingly, these global differences extend beyond urban versus rural, as seen when comparing the genus level gut flora profiles of children from rural Masi-Manimba to the unaffected children of the Kahemba LPZ (PERMANOVA $p = 3 \times 10^{-5}$) (Fig. 3e) and HPZ (PERMANOVA $p = 0.00321$) (Fig. 3f). When assessing specific differences in relative abundance at the genus level, we find that compared to Kinshasa, the unaffected children of the Kahemba HPZ harbors the most significantly different genera at 285, followed by Masi-Manimba with 215, while 137 genera were significantly different when compared to unaffected children in the Kahemba LPZ (expected BH-corrected p-value < 0.01, Wilcoxon test, ALDEx2) (Supplementary Data 3). To determine if study groups differed in potential functionality, we assessed the relative abundance of KEGG Orthology (KO) identifiers (Supplementary Data 5) using the Bray-Curtis index, and determined that like bacterial profiles, the functional profiles of these urban and rural groups also significantly segregated on global

measures (Supplementary Fig. 3). Again, like we observed when comparing differences in bacterial genera abundance, Kinshasa as compared to the HPZ of Kahemba harbored the most significant pairwise differences, with 446 KO's showing significant differences in overall relative abundance (BH-Corrected MWW, FDR < 0.01, Supplementary Data 6). While 137 genera were significantly different between Kinshasa and the LPZ of Kahemba, at a functional level this comparison yielded 312 KO's that had significantly different relative abundance (Supplementary Data 6). Despite large differences in bacterial genera abundance, Masi-Manimba compared to Kinshasa yielded the fewest differences in functional potential with 211 KO's reaching statistical significance (Supplementary Data 6). While functional differences between a rural and urban context are expected, surprisingly the functional differences observed between the two rural areas of Masi-Manimba and Kahemba are even larger (Supplementary Data 6).

**Machine learning accurately distinguishes populations**. To determine if bacterial gut flora of individual groups were distinguishable, we implemented random forest (RF) classifiers to

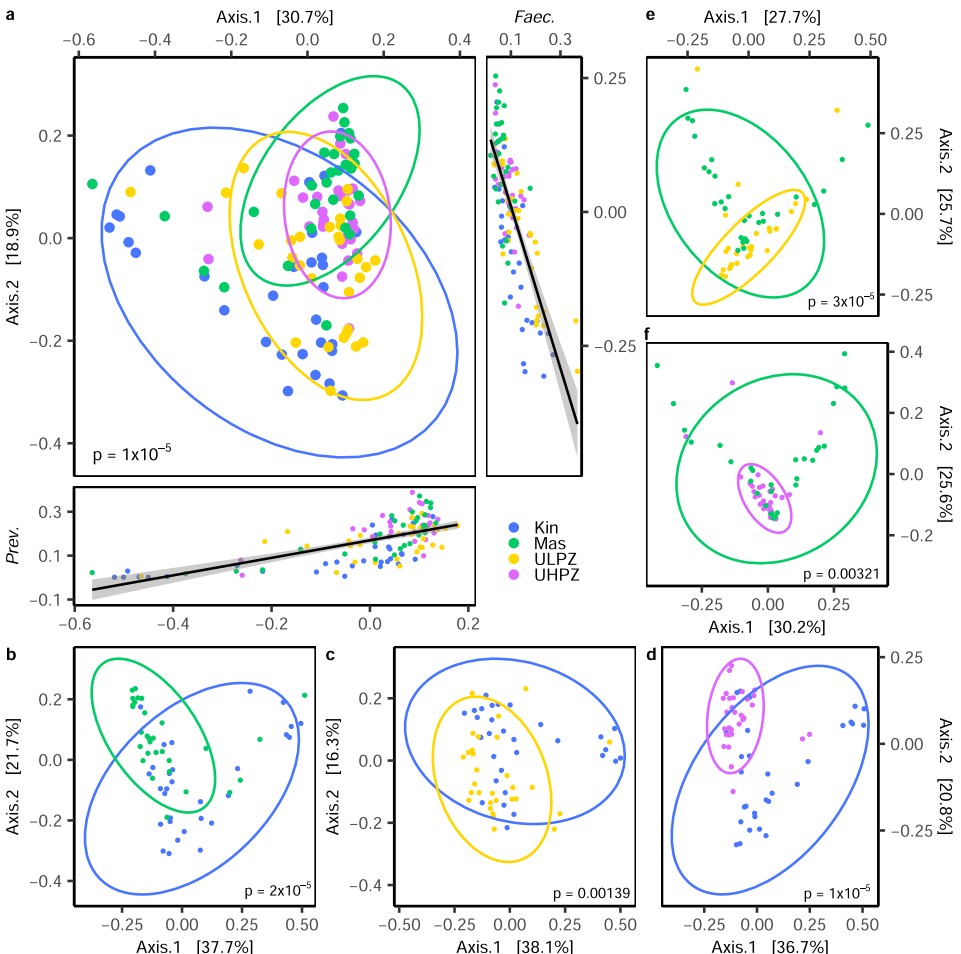

**Fig. 3 Global measure of gut bacteria dissimilarity at the genus level for a geographic context.** PCoA representations based on Bray-Curtis dissimilarity matrix values at the genus taxonomic level for **a** Kinshasa (Kin) vs. Masi-Manimba (Mas) and unaffected children from the low prevalence zone (ULPZ) and high prevalence zone (UHPZ) of Kahemba combined, **b** Kinshasa vs. Masi-Manimba, **c** Kinshasa vs. ULPZ, **d** Kinshasa vs. UHPZ, **e** Masi-Manimba vs. ULPZ, and **f** Masi-Manimba vs. UHPZ. Correlations in **a** were generated using Spearman's Correlation method of genus relative abundance against principal coordinate 1 and 2 axis values for each sample, and standard error with a 0.95 confidence interval is shown in gray with the regression line. Statistics for Bray-Curtis dissimilarity were generated using PERMANOVA.

evaluate whether machine learning algorithms could accurately classify study samples based on bacterial relative abundance profiles at the genus taxonomic level. We built six one-versus-all binary classifiers to classify samples from one geographic location compared to the rest (see Methods). Our classification models performed well in predicting samples across geographic locations (Fig. 4a, b), where, given the area under the receiver operating characteristic (ROC) curve, or the AUC, is 0.5 for a random classifier, the average AUCs for our models are 0.94 for Kinshasa, 0.89 for Masi-Manimba, and 87.3 for unaffected individuals from Kahemba LPZ or HPZ. The RF classifier performed well in distinguishing samples from Masi-Manimba compared to Kinshasa and unaffected children from the Kahemba LPZ with an AUC of 0.95 and specificity of 94% (Fig. 4a). The model was also very accurate at classifying the Kinshasa samples from the rest of the study groups, with AUC of 0.92 and 92% specificity, as well as the unaffected children from the Kahemba LPZ with AUC of 0.90 and 96% specificity (Fig. 4a). While highly accurate in distinguishing samples from urban and rural settings, the top ten most important genera that contributed to these distinctions varied by population location. The top three most important genera that distinguished Kinshasa from unaffected children in the Kahemba LPZ or Masi-Manimba for the classifier were *Actinomyces*,

*Clostridioides*, and *Leuconostoc*. Additionally, the relative abundance of all three of these genera were also significantly different between the groups in applicable pairwise comparisons (expected BH-corrected *p*-value < 0.01, Wilcoxon test, ALDEx2) (Supplementary Fig. 4) (Supplementary Data 3). When distinguishing Masi-Manimba from Kinshasa or unaffected children in the LPZ of Kahemba, *Phoenicibacter*, *Tolumonas* and *Rothia* were the top three most important features, with *Phoenicibacter* and *Tolumonas* being significantly different among groups in pairwise measures (expected BH-corrected *p*-value < 0.01, Wilcoxon test, ALDEx2) (Supplementary Fig. 4). *Denitrobacterium*, *Gemmatimonas*, and *Pandoraea* were the three most important RF features that distinguished the samples from Kahemba LPZ when compared to either Kinshasa or Masi-Manimba. The relative abundance of *Denitrobacterium* and *Gemmatimonas* were only significantly different between ULPZ and Mas (expected BH-corrected *p*-value < 0.01, Wilcoxon test, ALDEx2), while *Pandoraea* and *Denitrobacterium* were significantly different in relative abundance between ULPZ and Kinshasa (expected BH-corrected *p*-value < 0.01, Wilcoxon test, ALDEx2) (Supplementary Fig. 4). RF classifiers performed the best with highest overall prediction metrics for classifying samples from Kinshasa compared to those from Masi-Manimba and the unaffected children

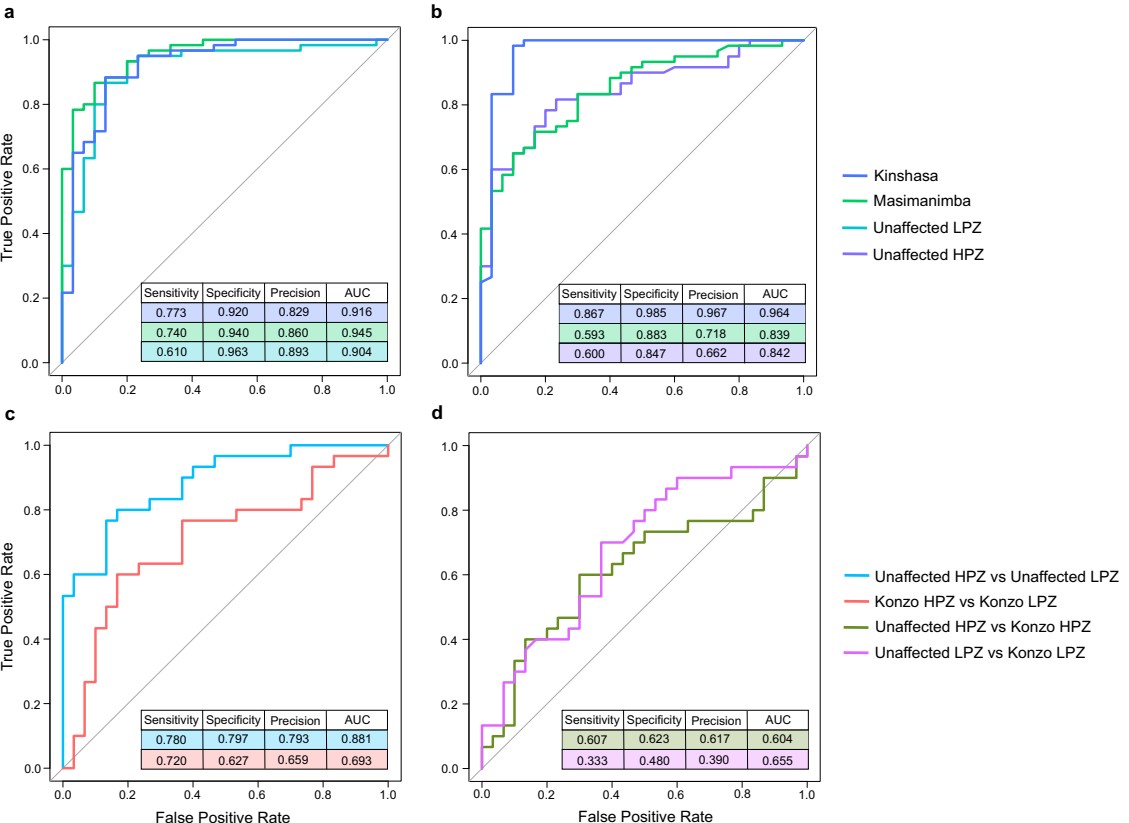

**Fig. 4 Random forest classification across populations.** Receiver operating characteristic (ROC) curves and classification performance metrics for one-vs-all random forest classifiers for **a** Kinshasa vs Masi-manimba vs Kahemba unaffected low prevalence zone (LPZ), and **b** Kinshasa vs Masi-manimba vs Kahemba unaffected high prevalence zone (HPZ), binary classifier for **c** unaffected individuals from HPZ vs unaffected individuals from LPZ, and those with konzo from HPZ vs konzo from LPZ, and **d** konzo vs unaffected individuals from LPZ and HPZ. All ROC curves and performance metrics are averaged over 10 repetitions of 10-fold cross-validation.

from the HPZ of Kahemba (Fig. 4b). Overall, the predictions from our RF classifier agrees with conclusions drawn from analysis using the Bray-Curtis dissimilarity index (Fig. 3a), adding additional confidence to suggest that the gut bacterial profiles are significantly different and distinguishable in an urban versus rural context as well as between rural regions of the DRC.

**Unaffected adolescents from konzo prone villages display markedly different gut flora profiles, but not functional capacity.** Given the strong differences in gut microbiome profiles observed across regions, we investigated whether distinguishable differences were also present between the two konzo prone villages within the same geographic region of Kahemba. When assessing the gut flora of the unaffected adolescents from the low konzo prevalence zone (ULPZ) compared to those unaffected from the high konzo prevalence zone (UHPZ), we observe significant segregation at the genus level based on Bray-Curtis measures (Fig. 5a) (PERMANOVA $p = 0.00057$), despite both groups having an overall similar diet, lifestyle, geographic setting, and chronic reliance on toxic cassava. Of the 494 bacterial genera that passed the ≥0.01% relative abundance in at least one of the six study groups, 63 were significantly different between the unaffected children residing in these two konzo prone areas, with the vast majority of these genera displaying higher abundance in unaffected children from the HPZ (expected BH-corrected $p$-value < 0.05, Wilcoxon test, ALDEx2) (Supplementary Data 2 and 3). Two highly abundant genera were most significantly associated with the principal axes of the Bray-Curtis dissimilarity ordination matrix, with *Faecalibacterium*, a butyrate-producing microbe, correlating most strongly with PCoA

Axis.1 (Spearman $\rho = -0.80$, $p = 5 \times 10^{-6}$) while *Prevotella* was most strongly correlated with PCoA Axis.2 values (Spearman $\rho = -0.93$, $p = 1 \times 10^{-9}$). The relative abundance of *Faecalibacterium* was also unexpectedly different, with the unaffected children from the HPZ harboring on average ~8% versus ~15% in those children from the LPZ (expected BH-corrected $p = 0.0078$, Wilcoxon test, ALDEx2) (Fig. 5b). However, when considering the compositionality of the dataset, the genus *Prevotella* fails to reach statistical significance, despite the relative abundance appearing largely different between these two groups. The random forest classifier was also able to distinguish the unaffected children from either the LPZ or HPZ at the bacterial genus level, with an AUC of 0.88 and 80% specificity (Fig. 4c). Lower abundance genera contributed most to the RF classifiers output, with *Gordonibacter*, *Denitrobacterium*, and *Tumebacillus* being the top three of the 10 most important features (Supplementary Fig. 5). While measurable differences in overall gut bacteria relative abundance were observed between these two groups, at the functional level, no differences were observed in pairwise measures of relative abundance of KEGG Ortholog (KO) identifications (MWW BH-Corrected FDR < 0.01) (Supplementary Data 6) or on a global measure of differences in KO distribution using the Bray-Curtis index (PERMANOVA $p = 0.05741$) (Supplementary Fig. 6a). Collectively, these data indicate that despite similar levels of nutritional deficiency, lifestyles, and diets high in cyanogenic cassava, the relative abundance of gut flora in unaffected children from these two konzo prone areas are significantly distinguishable, however it appears that the functional potential of the gut bacteria of both populations are similar overall.

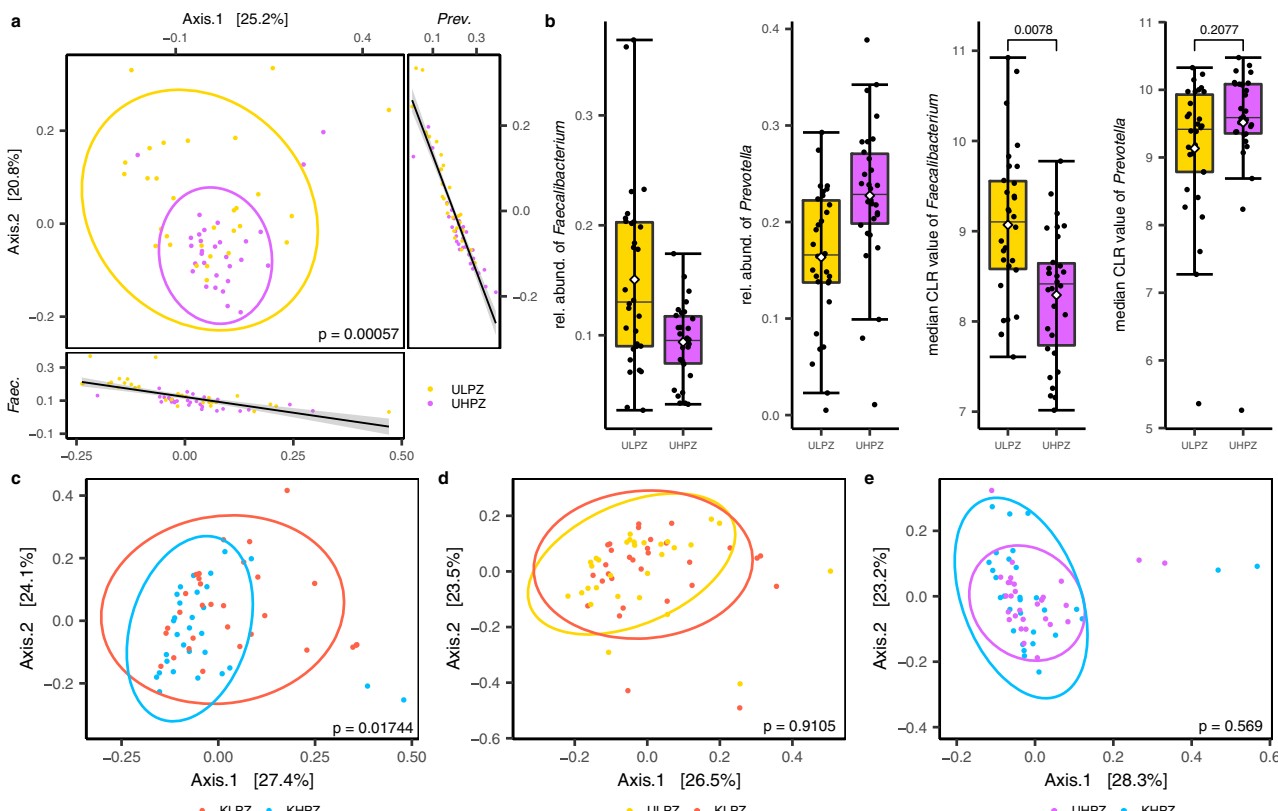

**Fig. 5 Global measures of gut bacteria dissimilarity at the genus level for the Kahemba region.** PCoA representations based on Bray-Curtis dissimilarity matrix values at the genus taxonomic level for **a** Unaffected children from the low prevalence zone (ULPZ) vs. Unaffected children from the high prevalence zone (UHPZ); correlations were generated using Spearman's Correlation method of genus relative abundance against principal coordinate 1 and 2 axis values for each sample, and standard error with a 0.95 confidence interval is shown in gray with the regression line **b** Distribution of the relative abundance and normalized CLR medians of both *Prevotella* and *Faecalibacterium* genera between unaffected children from the LPZ ($n = 30$) and HPZ ($n = 30$). Data are represented as boxplots where the diamond denotes the mean, middle line in the box is the median, the lower hinge is the first quartile, the upper hinge is the third quartile, and the whiskers extend from the lower and upper hinges to the smallest and largest value, respectively, at most to 1.5 * IQR (IQR, interquartile range, is the distance between the first and third quartile), with each individual value plotted. **c** PCoA representations based on Bray-Curtis dissimilarity matrix values at the genus taxonomic level for Konzo-affected children from the low prevalence zone (KLPZ) vs. Konzo-affected children from the high prevalence zone (KHPZ). PCoA representations based on Bray-Curtis dissimilarity matrix values at the genus taxonomic level for **d** Unaffected children from the LPZ vs. konzo-affected children from the LPZ of Kahemba and **e** Unaffected children from the HPZ vs. konzo-affected children from the HPZ of Kahemba. Statistics for Bray-Curtis dissimilarity were generated using PERMANOVA.

**Differences in the gut flora composition and potential functionality between konzo prone areas is less distinct for those stricken with the disease.** After establishing that notable bacterial abundance differences existed between unaffected children depending on the village of habitation in the Kahemba region, we next sought to determine if these differences were also observed between those affected with konzo. When globally comparing the gut flora profiles of affected individuals from the LPZ to those from the HPZ, we find that these populations significantly segregate based on Bray-Curtis dissimilarity measures (Fig. 5c) (PERMANOVA $p = 0.01744$). While statistically significant, the effect appears to be less pronounced than for those not affected with the disease (Fig. 5a). However, in pairwise assessments, the only 4 genera, that were significantly different in normalized abundance between individuals with konzo from these two zones were *Adlercreutzia*, *Slackia*, *Eggerthella* and *Gordonibacter*, (expected BH-corrected $p$-value $< 0.05$, Wilcoxon test, ALDEx2) (Supplementary Data 3). The minimal differences in genera abundance between children in a disease state from the LPZ and HPZ extends to functionality as well. Statistically significant differences were observed neither globally when comparing the relative abundance of KO identifiers that were ≥0.01% in at least one of the study groups using the Bray-Curtis index

(PERMANOVA $p = 0.053$) (Supplementary Fig. 6b) nor in specific pairwise comparisons of relative abundance of KO identifiers (BH-Corrected MWW, FDR ≤ 0.01) (Supplementary Data 6). Additionally, the random forest classifier also performed poorly when classifying these konzo-affected individuals from either the HPZ or LPZ based on genus level assessments, with an AUC of 0.69 and 63% specificity (Fig. 4c), whereas the classifier was more accurate in determining unaffected individuals from the same corresponding villages (Fig. 4c) (Supplementary Fig. 5). Taken together, this further highlights the notion of more bacterial similarity between individuals in a diseased state than between those without konzo, for reasons that remain elusive.

**The gut bacterial profiles between unaffected and konzo-affected individuals in their respected villages are indistinguishable.** Having established minimal differences between high prevalence and low prevalence zones based on a diseased or unaffected state, we next sought to assess if measurable differences were observable between cases of konzo and unaffected individuals within each prevalence zone. From a global view using the Bray-Curtis index, unaffected adolescents compared to konzo cases from their respected LPZ or HPZ villages do not segregate at the genus level (PERMANOVA $p = 0.9105$, 0.569, respectively) (Fig. 5d and 5e).

This trend was also observed in pairwise comparisons (using ALDEx2) on CLR transformed values, where zero statistical differences were observed using an FDR < 0.01 between individuals in a diseased state compared to the unaffected group from their corresponding village (Supplementary Data 3). As expected, there were also no measurable differences observed in relative abundance of KO identifiers that passed the filter criterion on both a global scale using Bray-Curtis index (Supplementary Fig. 6c and 6d) or in appropriate pairwise comparisons (MWW) using an FDR ≤ 0.01 (Supplementary Data 6). The random forest classifier also performed the worst with an average AUC of 0.63 for comparisons of konzo cases and unaffected children from their respected areas, stemming from the high degree of similarity between these groups, which was also recognized by all tested measures (Fig. 4d). Collectively, it appears that the gut flora of those with konzo compared to those who are unaffected from Kahemba are nonsignificantly different on all measures tested, indicating that if the microbiome is a modulating factor in the development of konzo, then the dietary practices and nutritional status of the Kahemba region likely puts the entire population of children at risk.

**Kahemba and Masi-Manimba harbor enrichments of gut bacteria and functional potential to exacerbate or moderate the effects of cyanogenic glucoside exposure respectively.** Given the high degree of similarity in gut flora structure between individuals with konzo compared to unaffected individuals from the Kahemba region, we next sought to determine if bacteria with varying degrees of documented linamarase/β-D-glucosidase activity were enriched in this region of the DRC. In pairwise comparisons that passed the abundance filter for analysis, we observe several bacterial species with known linamarase activities[31–34] that are significantly more abundant in children from both the LPZ and HPZ, regardless of disease status (Fig. 6a and Supplementary Fig. 7) (Supplementary Data 3). Two particular lactic acid/fermenting species, with high levels of linamerase activity, *Lactobacillus plantarum* and *Lactococcus lactis*, are >2× more abundant in both affected and unaffected children residing in Kahemba as compared to children of Kinshasa (expected BH-corrected *p*-value < 0.05, Wilcoxon test, ALDEx2, for all comparisons), while differences in the lower abundant *Leuconostoc mesenteroides* are less dramatic (Supplementary Data 2 and 3). When assessing differences from a konzo prone regions (Kahemba HPZ and LPZ) versus non-konzo regions (Masi-Manimba and Kinshasa) all three species are significantly different and enriched in Kahemba, with the most enzymatically active species, *L. plantarum* and *L Lactis*, showing the strongest differences in these comparisons (Supplementary Fig. 7). Interestingly, neither of these lactic acid bacteria show significant enrichment in children of Masi-Manimba as compared to Kinshasa, indicating that these observations are not exclusively an effect of urban versus rural differences (Supplementary Fig. 7) (Supplementary Data 3). While these LABs have been shown to biochemically possess the functional requirements to hydrolyze linamarin, the primary enzyme required, β-D-glucosidase, is not restricted to just those bacteria. Given that, we next sought to determine if sequences that mapped to β-D-glucosidase (EC: 3.2.1.21) (KO 5350) genes were also enriched in Kahemba. Interestingly, we observe that when compared to Masi-Manimba, a village whose diet most closely resembles that of Kahemba, genes that code for β-D-glucosidase (EC: 3.2.1.21) are enriched in unaffected and konzo-affected children from both the LPZ (BH-Corrected MWW $p = 0.013$, $p = 0.028$, respectively) and HPZ (BH-Corrected MWW $p = 0.034$, $p = 0.078$, respectively) (Fig. 6b) (Supplementary Data 6). While some bacteria harbor the potential to exacerbate the effects of linamarin exposure by harboring β-D-glucosidase enzymes, other bacteria have been shown to harbor the ability to detoxify cyanogenic compounds via pathways utilizing thiosulfate sulfurtransferase/

Rhodanese (EC: 2.8.1.1) and 3-mercaptopyruvate sulfurtransferase/MPST (EC: 2.8.2.1). When compared to the unaffected and konzo-affected children residing in the LPZ (MWW $p = 0.007$, $p = 0.016$, respectively) and HPZ (MWW $p = 0.008$, $p = 0.002$, respectively) of Kahemba, the children of Masi-Manimba on average have significantly more abundant representation of both bacterial MPST and Rhodanese genes (KO1011) (Fig. 6c) (Supplementary File 6). Collectively, these data highlight two plausible scenarios as to how the gut microbiome can modulate the development of konzo, through either a susceptibility or protective scenario, under the assumption that all other required factors are present that enable the development of konzo.

## Discussion

In recent years there has been much interest into investigating the gut microbiome structure of understudied populations, particularly individuals from the African continent to better understand how this symbiotic relationship varies across human populations[5,6]. Given the limited studies in this region, we sought to investigate the gut microbiome structure of individuals from the DRC, with a particular focus on children who are afflicted with cassava induced neurotoxicity/konzo. This multifactorial disease predominantly affects children and women of child-bearing age in sub-Saharan African countries including Tanzania, Cameroon, Mozambique, Central African Republic, and the DRC, particularly in Kahemba, Bandundu province[25,29]. The occurrence of konzo is strongly associated with the consumption of improperly processed bitter cassava coupled with malnutrition and environmental stressors such as drought and turbulent times, leading to irreversible spastic paralysis and neurocognitive deficits[35]. To understand how the gut microbiome may modulate disease occurrence, we used shotgun metagenomic sequencing to assess the gut flora profiles from unaffected and presumable healthy children residing in the urban capital of Kinshasa, a rural village with no documented history of konzo outbreaks, yet who rely on cassava as their staple diet, Masi-Manimba, and two areas with different konzo prevalence in the Kahemba region.

Initially, we evaluated the structure of the gut microbiome in relation to an urban versus rural context to establish a baseline of expectation from these regions of the DRC. We found that regardless of region, all study groups on average harbored >450 unique species with levels of α-diversity that were indicative of "diverse" microbiomes. While all groups appear to harbor diverse microbiome structure, numerous differences were detected when comparing the profiles of individuals from Kinshasa to those residing in either Masi-Manimba or Kahemba. On global measures at the genus taxonomic level, the urban population significantly segregates from both rural groups of children based on Bray-Curtis dissimilarity measures. Significant segregation of microbial profiles was also observed for the two different rural regions, indicating regional specifications and influences that contribute to the overall structure of the gut flora in this study population, outside of a simple urban versus rural context. These findings were further supported with the use of a random forest classifier that was also able to accurately distinguish these populations based on bacterial abundance profiles. Unique differences at the genus level were detected in pairwise assessments of bacterial relative abundance when comparing Kinshasa to rural sites; however, the vast majority of said differences were shared, highlighting specific genera that were consistently more or less abundant in urban or rural settings. We also observe trends in bacterial enrichment that have been traditionally associated with western-based diets versus diets of rural populations. Numerous studies have demonstrated that bacteria within the genus *Bacteroides* are associated and more abundant in humans that

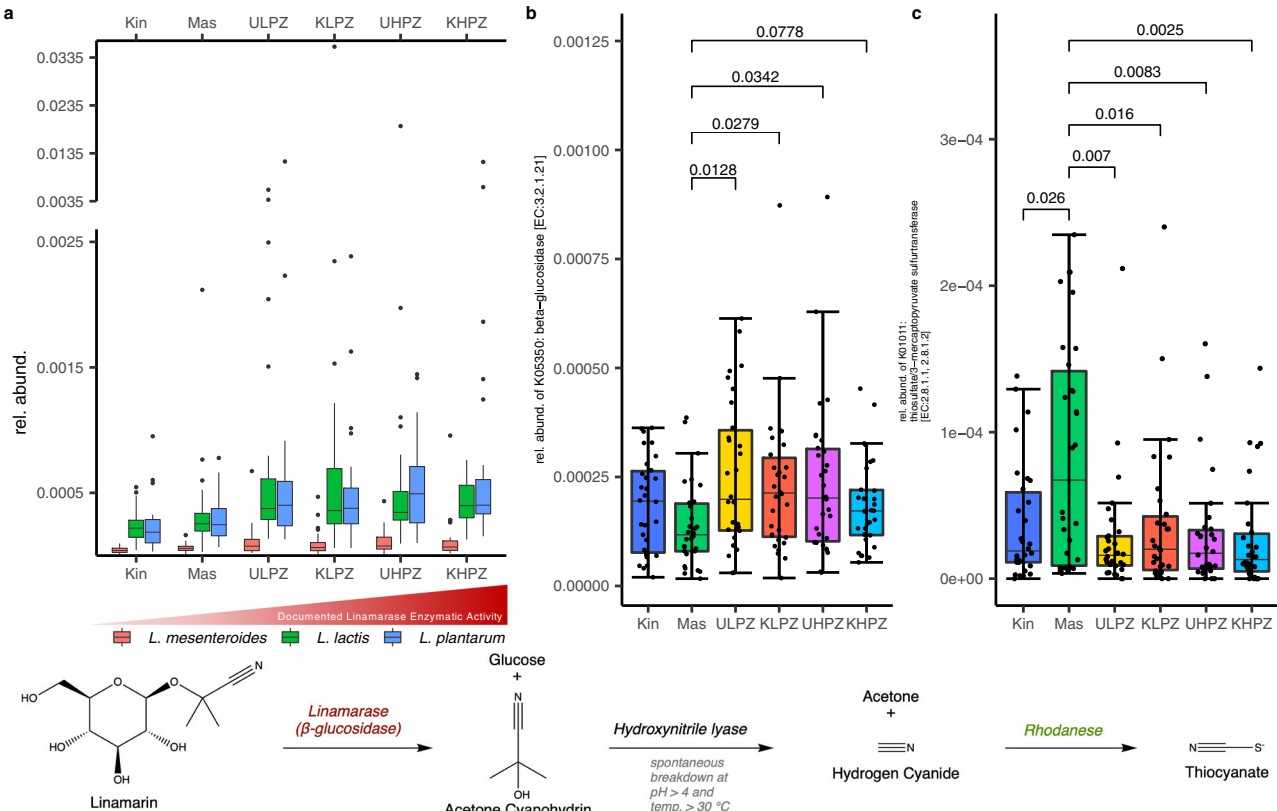

**Fig. 6 Abundance of relevant lactic acid bacteria, linamarase, and rhodanese in study populations. a** Boxplot distribution of relative abundance for L. *mesenteroides, L. plantarum,* and *L. lactis.* Statistics are based on pairwise comparisons using the Mann-Whitney-Wilcoxon test and reported as expected BH-corrected *p*-value FDR < .05, two-sided Wilcoxon test, ALDEx2 (Supplementary Fig. 7). **b** Distribution of the abundance of β-glucosidase (KO 5350) [EC.3.2.1. 21] between the 6 study groups. Statistics are based on pairwise comparisons using the two-sided Mann-Whitney-Wilcoxon test. **c** Distribution of the abundance of Rhodanese/thiosulfate/3-mercaptopyruvate sulfurtransferase (KO1011) [EC. 2.8.1.1/2.8.1.2] between the six study groups. Statistics are based on pairwise comparisons using the two-sided Mann-Whitney-Wilcoxon test. In **a–c**, samples are from Kinshasa (Kin) (n = 30), Masi-Manimba (Mas) (n = 30), Unaffected Low Prevalence Zone (ULPZ) (n = 30), Konzo Low Prevalence Zone (KLPZ) (n = 30), Unaffected High Prevalence Zone (UHPZ) (n = 30), and Konzo High Prevalence Zone (KHPZ) (n = 30). Additionally, data are represented as boxplots where the middle line in the box is the median, the lower hinge is the first quartile, the upper hinge is the third quartile, and the whiskers extend from the lower and upper hinges to the smallest and largest value, respectively, at most to 1.5 * IQR (IQR, interquartile range, is the distance between the first and third quartile). In **a**, outliers are plotted individually, and in **b** and **c**, each individual value is plotted.

consume western style diets rich in refined sugar, carbohydrates, and fat, whereas enrichment of bacteria from genus *Prevotella* are most frequently associated with diets rich in fiber and unprocessed foods[5,22,36]. Interestingly, within our study population, we observe similar trends showing children residing in rural locations have significantly higher representation of *Prevotella* and children from Kinshasa having a microbiome dominated by *Bacteroides*. The vast majority of studies that associate such findings often compare industrialized western countries to rural sampling sites; however, here we observe the same trends within a single country of origin, as was also observed in studies using 16 S sequencing of gut flora focusing from individuals of Nigerian[37] and Himalayan descents[38]. While Kinshasa is an urban city, we would not consider the diet western per se, therefore these associations observed in this study are most likely influenced by the higher degree of dietary diversity for residents of Kinshasa compared to less diversification in rural DRC. Collectively, these findings add to a growing body of literature investigating the gut flora of non-westernized regions and highlight key differences between those individuals residing in the urban capital of Kinshasa versus those who live in rural villages of the Congo.

The DRC represents a unique region to investigate the intersection of diet and the microbiome, as particular regions such as

Kahemba rely solely on a monotonous diet of bitter cassava, with very little protein intake. This dietary combination coupled with malnourishment of those who reside in the region, notably a severe lack of sulfur amino acids, sets the stage for susceptibility to the development of konzo[25,39]. While the dietary factors known to cause konzo are for the most part uniform in Kahemba, the prevalence of this disease is variable between villages, but can be as high as 10% of the population[25]. Additional factors and their contribution to enabling the development of konzo remain unclear; however, underlying putative gene and environmental interactions, or as it pertains here, gut microbial components have been speculated[25]. When genus level gut bacterial profiles were compared from unaffected children residing in a zone of high prevalence versus unaffected children from a zone of lower prevalence using the Bray-Curtis index, significant segregation was observed. In direct pairwise comparisons of bacterial abundance between these groups, 63 genera showed significant differences. Notably, large differences were observed in dominating genera such as *Faecalibacterium*, an unexpected finding considering the general homogeneity of lifestyle, dietary practices and high levels of malnourishment of these two study populations. The random forest classifier was again accurate in distinguishing these two populations at the genus level with an AUC of 0.88,

performing nearly as well as detecting urban or rural groups. Interestingly, when children affected with konzo from these two villages were compared, we still observe significant profile segregation based on Bray-Curtis measures; however, there were no statistically significant differences in direct pairwise comparisons of bacterial abundance between these groups. This lack of difference conclusion was also supported using the random forest classifier, which was not very accurate in distinguishing these two groups, particularly compared to the accuracy when distinguishing unaffected individuals from their respective high or low konzo prevalence villages. Collectively, these findings suggest that individuals in a diseased state have an overall more similar microbiome than those who are unaffected between the two villages of study. While we cannot determine from these data if the microbiome of these individuals were the same prior to the onset of konzo or if having konzo is what contributed to the striking similarity between these individuals. However, having konzo limits ones mobility, reduces social engagement due to disability and contributes to stigmatization, all of which likely influence overall diet, environmental exposures (via limited mobility), and normal activity, all possible contributors that shape similarities in the microbiome profiles of these individuals.

In searching for bacterial differences between unaffected and those with konzo within their respected villages, we observe no statistical segregation of bacterial profiles at genus level assessments using the Bray-Curtis index. Additionally, the random forest classifier performed the worst in these comparisons, further supporting the conclusion that both affected and unaffected adolescents from the same villages are virtually indistinguishable, on global measures.

Given the high degree of bacterial similarity between children with konzo compared to unaffected individuals, our data suggest that if the microbiome contributes to the development of konzo, then perhaps the entire region of Kahemba is at risk, as recently inferred[40]. This is a plausible notion considering the unaffected children in this region are by no means "healthy", as they too are in a state of malnourishment and chronically rely on improperly processed cyanogenic cassava as their main source of food; the key risk factors for developing konzo. Given the monotonous consumption of cassava as the staple for the Kahemba population, we sought to determine if bacteria with known linamarase activity were enriched in these populations as a whole. To our surprise, we identify several species of lactic acid bacteria that were significantly more abundant in the Kahemba region regardless of disease status, particularly when compared to children of Kinshasa. Notably, the relative abundance of both *Lactobacillus plantarum* and *Lactococcus lactis* is more than doubled in the gut microbiomes of the children of Kahemba compared to those of Masi-Manimba and tripled when compared to Kinshasa. Other lactic acid species such as *Leuconostoc mesenteroides* were also significantly more abundant in the Kahemba region than in the children of Masi-Manimba and Kinshasa. These findings are of interest as these particular bacterial taxa have been demonstrated as key facilitators of cassava fermentation, and monotonous consumption of these foods as is the case in Kahemba, could potentially elevate their abundances within the gut microbiome, be it transitory or permanent[31,33,34]. Traditionally, lactic acid bacteria are considered "pro-biotics" and beneficial for a healthy gut microbiome[41]; however, in the case of konzo, their enrichment may represent a cautionary tale.

Linamarin, cassava's primary cyanogenic glucoside cannot be directly utilized for energy by humans, and if ingested should typically be secreted intact through urine. However, if hydrolyzed in the digestive tract by resident bacteria that possess the required β-D-glucosidase (EC: 3.2.1.21) enzyme, results in the release of the glucose and acetone cyanohydrin molecules, leading to toxicity[42]. As β-D-glucosidase is not exclusively restricted to lactic acid

bacteria, we sought to determine if functional genes that code for this enzyme were also enriched in the Kahemba region. When compared to Masi-Manimba, a village whose diet and living environment is more similar to that of Kahemba, yet outbreaks of konzo have not been identified, we see also significant enrichment of genes that code for β-D-glucosidase (EC: 3.2.1.21) (KEGG ortholog 5350). Collectively, it appears that children of Kahemba not only harbor enrichments of bacteria that have been demonstrated to hydrolyze linamarin, but also contain an overall higher abundance of genes that code for β-D-glucosidase, when compared to a village of similar structure. While the diet and living conditions of Masi-Manimba are similar to that of Kahemba, why outbreaks of konzo do not exist in that region remains unknown. However, when assessing genes that could serve to detoxify cyanogenic compounds, we see that the children of Masi-Manimba on average have higher abundance of both bacterial thiosulfate sulfurtransferase/Rhodanese (EC: 2.8.1.1) and 3-mercaptopyruvate sulfurtransferase/MPST (EC: 2.8.2.1) (KEGG ortholog 1011) as compared to the children of Kahemba. These data highlight a scenario where bacterial abundance and functional genes could exacerbate the release of cyanide after ingesting cyanogenic glucosides in the children of Kahemba, as well as a possible scenario of added protection/detoxification in Masi-Manimba.

While this study is the first investigation into the gut microbiome of children that rely on a monotonous cyanogenic rich diet, the notion of the involvement of gut bacteria in hydrolyzing linamarin and other cyanogenic glucosides in the guts of the host are not novel. Studies from the early 1970–1990's demonstrated that preparations derived from rodent caecal material and bovine ruminal contents possessed the biochemical ability to liberate cyanide from not only linamarin, but other relevant cyanogenic sugars such as amygdalin and prunasin[43–45]. It has also been shown that amygdalin, a former cancer remedy as well as a cyanogenic glucoside found in almonds, is nonlethal if orally administered to germ-free mice[46]. However, if the same dose is given to mice colonized with bacteria, it can result in lethality[46]. Collectively, the involvement of the gut microbiome in relation to liberating cyanide derivatives has been established; however, more recent data are scarce, particularly in a human context. Conversely other mammalian species that frequently ingest toxic compounds in their food have developed a gut microbiome that serves to aid in detoxification of these substances, as has been observed for the bamboo eating Panda Bear[26] and most notably the creosote eating desert wood rat[27,28]. While our findings indicate enrichments of bacteria capable of hydrolyzing linamarin as well as genes coding for β-D-glucosidase in the children of Kahemba, the development of konzo is multifactorial in nature with numerous environmental variables and stressors. As such, the gut microbiome cannot be the sole cause of disease, but rather a required modulator, as without a functioning gut microbiome, linamarin, and other cyanogenic glucosides would pose little to no risk to humans. With additional investigation, components of the gut flora may serve as targets to mitigate the susceptibility of konzo in the DRC and other vulnerable populations around the globe; a subject of global health relevance, as reliance on cassava and its food products will continue to rise as populations expand and agricultural environments change.

## Methods

**Sample collection.** During March of 2018, our research group comprised of DRC based physicians and experts on konzo along with research scientists collected 180 stool samples and 7-day dietary recall questionnaires from study populations in Kinshasa, Masi-Manimba, and Kahemba, DRC. Prior to collection, the Ministry of Health for the DRC and the institutional review board at the Oregon Health and Sciences University provided ethical approval for this study. All participants and parents were consented prior to collection in either French or the appropriate language for the region of collection. Stool samples were self-collected from all participants then transferred to cryovials and stored in liquid nitrogen within 1 h of

sample collection, by our research team. Stool samples stored in nitrogen collected outside of Kinshasa were transported back laboratories at the INRB in Kinshasa prior to cold chain shipment to the USA for sample preparation and sequencing. During sample collection an assessment as to whether an individual was affected with konzo was conducted following the WHO's 3 main criteria for diagnosis including evidence of a (1) visible symmetric spastic abnormality of gait while walking or running, (2) a history of onset of less than 1 week followed by a nonprogressive course in a formerly healthy person, and (3) bilaterally exaggerated knee or ankle jerks without signs of disease of the spine[47,48].

**DNA extraction, quantification, and sequencing**. Total DNA was extracted from ~250 mg of stool sample for each individual using the QIAmp PowerFecal DNA Kit (Qiagen) following manufacturer's protocol and quantified using the Qubit dsDNA BR Assay Kit (ThermoFisher Scientific). DNA was then stored at −20 °C prior to sequencing. DNA was submitted to the Genomics Core at George Washington University for shotgun metagenomic sequencing. Sequencing libraries were constructed using Illumnia's Nextera XT kit following manufactures protocol and sequenced in 3 runs on the NextSeq500 High-Output to increase read depth per sample. An average of 5,288,982 (sd = 1310988.0) total reads were assigned from Kinshasa, 6,089,750 (sd = 310897.1) from Masi-Manimba, 6,493,479 (sd = 1089570.3) from ULPZ, 6,276,254 (sd = 735609.8) from KLPZ, 5,906,960 (sd = 575714.1) from UHPZ, and 6,512,447 (sd = 396276.4) from KHPZ.

**Determination of the bacterial composition**. We trimmed Illumina adapter sequences and removed low-quality base-pairs from the metagenomic reads using skewer (v0.2.1). Potential human host reads were filtered out using BMTagger (v3.101) by aligning reads to the human reference genome, hg38 (UCSC), prior to microbial abundance estimation. Kraken 2 (v2.0.6)[49,50] and Bracken (v2.0.0)[51] were used to assign DNA sequences to taxonomic labels and to compute species abundance. The standard Kraken 2 databases (human, bacteria, viral, and archaea) were used for both Kraken 2 and Bracken. We processed Kraken/Bracken outputs with Pavian[52] and carried out all downstream statistical analyses and data visualization in R Studio (v3.6.1). Read counts for each taxonomic classification were converted to relative abundance within each sample to account for the differences in sequencing depth. We determined the alpha and beta diversities using R Packages, phyloseq (v1.28.0)[53] and vegan (v2.5-6). We used number of species and the Shannon Diversity Index to estimate alpha diversity, and the Bray-Curtis dissimilarity matrix and principal coordinates analysis (PCoA) to estimate and visualize beta diversity in the samples. To better estimate species richness in the sample and remove likely superfluous low abundance taxa, all species with a relative abundance less than 0.01% for a sample were set to zero, only for calculating number of species and the Shannon Diversity Index. Prior to assessing beta diversity and determining significantly different taxa between specific pairwise comparisons (further explained in Statistical Analysis section), taxa with low relative abundance were removed, but using an overall filtration method that was also used for KO data. Taxa (or KOs when applicable) that on average had a relative abundance greater than or equal to 0.01% in any group were retained for further analysis.

**Determination of functional annotation**. We removed Illumina adapter sequences and performed quality trimming using FASTP (v0.20.0)[54] using default parameters (minimum base quality: 15, max number of "N" bases in a read: 5, polyG trimming). The resulting reads were annotated using Kraken 2 with database consisting of RefSeq bacterial, archaeal, virus, fungi, and human[50], and the reads that were not identified as human were retained for further functional annotation. These reads were aligned against the KEGG microbial gene database[55] using bowtie2 (v2.4.4)[56] with default parameters except, secondary alignment was omitted (--omit-sec-seq). The alignment results were concatenated to KEGG orthologs KO using custom Perl scripts (https://doi.org/10.5281/zenodo.5171168) and KEGG relational tables. The resulting read counts were also converted to relative abundance prior to applying the same overall filtration done in the analysis of bacterial composition. Any KOs with an average relative abundance of greater than or equal to 0.01% in any one group were retained for further analysis.

**Random forest method**. We implemented random forest (RF) models using taxa summarized at the genus level. We filtered for rare genera by retaining only those taxa that are present at least 0.01% relative abundance in at least 25% of samples, resulting in 519 distinct taxa at the genus level used in the random forest model. We then applied centered log ratio (CLR) transform on the filtered taxa count matrix to account for compositionality effects. To compare between geographic locations (Kinshasa, Masi-manimba, and unaffected adolescents from two villages in Kahemba, HPZ and LPZ), we used binary classification approach and built six one-versus-all binary RF classifiers to classify samples from one geographic location compared to the rest. We also built binary classifiers for classifying between unaffected individuals from HPZ versus unaffected individuals from LPZ, konzo individuals from HPZ versus konzo individuals from LPZ, konzo versus unaffected individuals within HPZ and within LPZ. To build these models, we performed 10 rounds of 10-fold cross-validation (using R package caret), using accuracy as the metric for selecting the optimal model. The performance metrics and ROC curves were averaged across the cross-validation rounds. The ROC curves and

performance metrics showing sensitivity-specificity trade-off and classification performance for each classifier are shown in Fig. 4a–d.

**Statistical analysis**. Alpha diversity measurements were determined using the estimate richness function of the phyloseq package[53]. To test the statistical significance for the difference in beta diversity (ex: Bray-Curtis Dissimilarity), PERMANOVA analysis using the adonis function with 99,999 permutations in R Studio was used on relative abundance values for genus that passed our overall filtration scheme. The initial test was done on the Bray-Curtis distance matrix for all 180 samples, using a formula incorporating factors geography, region, disease, age, and sex; the formula tested for each factor independently and any possible interactions. Furthermore, the adonis function was used to analyze variance using the Bray-Curtis dissimilarity matrix for relative abundance data for KOs as well. The results from the specific comparisons performed were visualized as PCoA plots using ordination. Using a Spearman Correlation, genus relative abundance was correlated with PCoA values for the corresponding axis 1 and 2 values to determine which bacterial genus associated with the principal coordinates. To account for compositionality, the ALDEx2 (v1.16.0) package in R studio was used to determine differences in taxa abundance between specific pairwise comparisons. The counts for taxa at each taxonomic rank that were retained after filtration were tested using default parameters for the aldex function (including mc.samples = 128, test = "t", denom = "all"). The aldex function takes in read counts as input and performs CLR transformation to infer abundance prior to performing statistical testing[57,58]. The expected Benjamini-Hochberg (eBH), FDR < 0.05, corrected p-value of the Wilcoxon test was used to determine differentially abundant taxa between different pairwise comparisons. Additionally, the Mann-Whitney-Wilcoxon (MWW) test was done as a post-hoc test corrected for multiple testing with a Benjamini-Hochberg correction of 0.01 FDR to determine specific differences in each of the presented pairwise comparisons for relative abundance differences for the various taxonomic classifications and KOs.

**Reporting summary**. Further information on research design is available in the Nature Research Reporting Summary linked to this article.

## Data availability
The raw FASTQ files generated from shotgun metagenomic sequencing have been deposited in NCBI's Sequence Read Archive (SRA) database under BioProject PRJNA752006 for open access. OCHA Humanitarian Data Exchange's datasets on DR Congo-Health Zones (https://data.humdata.org/dataset/dr-congo-health-0) and Malnutrition datasets (https://data.humdata.org/dataset/rdc-taux-de-la-malnutrition-decembre-2019) were used to generate the map in Fig. 1 and the datasets are free available to the public. The KEGG database (https://www.genome.jp/kegg/) was used as reference for identifying genes present in the dataset (Fig. 6). All additional data used in the reported findings have been made available in the Supplementary Data Files, with specific references when relevant in the manuscript.

## Code availability
The code used in this manuscript has been deposited in Zenodo from github (https://doi.org/10.5281/zenodo.5171168). Although the code used in this manuscript is not entirely custom and default parameters are used when utilizing the various software/packages, any deviations from the default settings have been noted in the manuscript.

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

## Acknowledgements

We would like to thank all of the Congolese participants of this study for kindly donating specimens for microbial analysis. We would also like to thank the funding sources for this project, with DTK being supported by NIH grant NIEHS/FIC R01ES019841, E.V. being supported by the A. James Clark Distinguished Professor of Molecular Genetics Endowment and M.S.B. being supported by the Fogarty International Center of the National Institutes of Health (NIH) under Award Number D43TW009343 and the University of California Global Health Institute (UCGHI); The content is solely the responsibility of the authors and does not necessarily represent the official views of the NIH or UCGHI. We would also to acknowledge the passing of a co-author, Jean-Pierre Banea Mayambu, a pioneer in the field of konzo, who will truly be missed.

## Author contributions

M.S.B. and N.V. conceived and designed this study along with data collection, processing, and analysis as well as manuscript preparation. A.K., S.P., A.M., R.B. and M.N.A.U significantly aided in data analysis, machine learning applications and graphic representations in this manuscript. D.S., C.M., J.N., K.K., H.M. and P.M. aided in sample collection, consent and questionnaires that were administered to participants of this study, as well as data analysis. B.C., B.M., M.B., A.D., D.O., D.M.-N., R.B., D.T.-K. and E.V. provided senior guidance as well as ethic approvals in the DRC and USA for this study, as well as providing oversight of study design, data collection, analysis, and final manuscript preparation. Both M.S.B. and N.V. contributed equally and have the right to list their name first when referencing this work.

## Competing interests

The authors declare no competing interests.

## Ethics statement

Prior to any specimen collection, community consent was first obtained from village leaders. Informed and written consent was then obtained from the Chef de zone/Médecin de zone, who represent the interests of the ministry of health and individuals in the study population. Upon approval and consent by the representatives, verbal and/or written consent was obtained from the parent and/or guardian of the children that participated in the study. Verbal consent was obtained when there were limitations with literacy and the individual expressed a general disinclination to signing written documents that cannot be read and fully comprehended by them. The study posed no harm to subjects, and participants could choose to not donate samples. The study was approved by the IRB review board at the Oregon Health & Science University (OSHU) (IRB FWA00000161) and from the Ministry of Health of the Democratic Republic of the Congo (DRC).
