## [Peer Review File · Nature Communications]

REVIEWER COMMENTS

Reviewer #1 (Remarks to the Author):

In this manuscript the authors describe the microbiome of Africans in DRC from an urban, rural, and two villages with a high incidence of Konzo. They find a difference in microbiota composition that is mostly dependent on geography and urbanization and very little difference in microbiota composition between individuals with Konzo and those as yet diagnosed.

It is difficult to assess many of the conclusions given the lack of details provided in the methods and the somewhat outdated analysis pipeline used.

Specifically, alpha diversity comparisons are highly sensitive to sequencing depth. The authors do not make clear whether any rarefaction took place in order to compare the α -diversity of the different groups. Furthermore, it does not appear that any trimming of low-abundance reads occurred for the α -diversity calculations. Given that the authors use OTUs in lieu of the more commonly accepted DADA2 for “species” it is highly likely that many of the OTUs are in fact spurious and may not represent true diversity. See Prodan et al, (2020) Comparing bioinformatic pipelines for microbial 16S rRNA amplicon sequencing and Callahan et al, (2017) Exact sequence variants should replace operational taxonomic units in marker-gene data analysis for why OTUs are problematic.

Presenting data as shown in Figures 2C-E is problematic since these graphs do not reveal associated error (nor is it mentioned in the text). Differences in relative abundance of phyla, families, and genera mentioned in the manuscript are difficult to interpret if the authors don't reveal the error associated with these percentages. The difference in Bacteroides/Prevotella ratio in the Supplement is the most convincing given the statistically analysis associated with those data and that the finding is in line with that reported now in several manuscripts across not only African populations but those in South America and Asia.

The number of genera that are reported to be different between groups is quite high (data associated with Fig 3) and likely a result of spurious OTUs. This analysis should be redone with ASVs to have a more reliable count of ASVs thereby making comparisons between groups more robust. It is also important to note that many differences described are from very low abundance taxa therefore it is incredibly important that care is taken to make sure these are in fact reliable “species”.

The authors conclusions from Fig7 are problematic in the absence of metagenomic data. The authors are assuming that sequenced strains from western equivalent taxa are similar enough to extrapolate the genomic content from strains coming from individuals from DRC. There is in fact evidence from comparisons of genomes from strains of *Prevotella* from western and non-western samples that this is not necessarily the case. See Tett et al., (2019) The *Prevotella copri* Complex Comprises Four Distinct Clades Underrepresented in Westernized Populations.

Similarly in the discussion there are several problematic conclusions.

First, the statement: “We showed that regardless of region, all study groups on average harbor >4000 mapped OTUs with high levels of α -diversity, indicative of “healthy” microbiomes.” Has several issues. Something can have a high level of diversity by comparison to something that is lower, however there is no comparison in this study. Given the sensitivity of α -diversity measures to read depth and how species are determined in this study, reporting a number for species and saying it is “high” is not appropriate. (Authors also refer to “high” alpha diversity within the main text at Line 121.) Second, the authors need to be very careful in equating “high” diversity and good health in a population for which this has not been studied. Associations between microbiota diversity and health have been exclusively performed in western populations. Given the significant difference in the microbiome of more traditional populations from that of westerners, one cannot assume that high diversity in the traditional population microbiota is also associated with better health.

The section entitled: “Gut bacteria capable of hydrolyzing linamarin are enriched in the Kahemba region” proposes that enrichment of organisms within the microbiota known to harbor linamerasases could release additional cyanide within the gut. This hypothesis is extremely premature given the data presented. First, it assumes that strains they detected also carry linamerasases. The studies they reference are from strains isolated from cassava not the human gut. Metagenomic sequencing or strain isolation and whole genome sequencing would be required to address whether there is in fact increase linamerasases within the microbiome of these participants or whether the strains of *Lactobaccillus*, *Lactococcus*, or *Leuconostoc* harbored within their microbiota do in fact have linamerase activity. Second, it assumes that linamerase activity within the gut is a major source of cyanide release. Linamerase activity could be measured within the stool samples to determine whether there is in fact elevated activity in those with Konzo. Measuring consumed amounts of linamarin would be important to rule out the simplest explanation, which is that those with Konzo are consuming more linamarin in general, perhaps from less fermented cassava. It is also possible that those with Konzo are lacking protective strains (assuming the microbiota is involved in onset or severity of this disease). In other words, the data presented here does not provide a compelling hypothesis to explain a microbiota link to Konzo.

Minor points:

What is the y-axis in Supplementary Figure 1? Servings? How is this normalized?

The colors in Fig 3 make it very hard to discern groups.

Figure 5 legend is confusing and doesn't refer to all panels.

Fig 7B is significant relative to Kin or Mas or both? For all 3 strains graphed? Need more clarity for these data.

Line 311: Authors report that *Leuconostoc mesenteroides* is more abundant in children from Masi-Manimba as compared to Kinshasa. Is there is difference in Konzo prevalence between these 2 areas?

Line 362: Jha et al., (2018) Gut microbiome transition across a lifestyle gradient in Himalaya. And Ayeni et al. (2018) Infant and Adult Gut Microbiome and Metabolome in Rural Bassa and Urban Settlers from Nigeria

Both demonstrated changes in the microbiota of people living on a lifestyle gradient within a single country of origin (including observing *Prevotella* to *Bacteroides* differences). Would be important to cite these studies as well as put the data presented here in the context of those prior papers.

Reviewer #2 (Remarks to the Author):

In this paper, Matthew Bramble, Neerja Vashist and colleagues investigate the association between the gut microbiome and Konzo, a neurodegenerative disease mostly affecting children and caused by the ingestion of food toxins. Konzo mostly occurs in low-income regions, and the authors sampled cohorts in different urban and rural regions in DRC, including from areas with no, low and high prevalence of Konzo.

The main strength of the paper is to present microbiome data from populations that are seriously under-characterized in microbiome science across a range of different factors: lifestyle (non-industrialized lifestyle), geography (DRC, and urban vs. rural), age (this is an infant cohort) and disease status (Konzo is under-studied, especially on its connection to the gut microbiome). The design of the cohort is thoughtful: the authors recruited participants in both urban and rural regions and in both low and high prevalence regions. In each of these sampling areas, a similar number of

participants (30) have been recruited. This recruitment scheme should open the possibility of teasing apart contributions of lifestyle, geography and disease status on inter-individual differences in the microbiome, which could make the study quite powerful.

However, I'm not convinced that the authors effectively disentangled the contribution of geography (and/or diet) from the contribution of disease on differences in microbiome compositions. No results currently presented in this paper support a potential association between the microbiome and Konzo. I am also not convinced that the authors have the necessary data types to discover such associations, if there are any.

Overall, the authors should use statistical techniques to account for multiple factors at once, and measure the effect size and p-value of each individual factor. Right now, the authors performed analyses separately for each factor, which prevents the accurate measurement of the real contribution of each variable, especially disease status. Factors should include disease status, geographic region of sampling, age, sex, diet, etc.

Section L262-275 and figure 5/6: the comparison between all unaffected and all affected children is lacking. This is the control experiment that we need to see whether individuals with disease state have higher microbiome similarity independently of geography of origin. Had the authors analyzed their data with models that handle the simultaneous effect of multiple variables, the measurement of the effect size of geography vs. diet vs. disease would have been made possible.

Another example of inappropriate dissection of the factors: Fig 7A does not show whether the relative abundance of *Turicibacter* is significantly higher in affected children of the low prevalence zone (KLPZ) compared to unaffected children of the high prevalence zone (UHPZ). This is necessary to exclude an effect of village/geography.

Fig. 7B is also another illustration of a strong effect of geography, and not disease. Many comparisons don't go the way expected if there was an effect of disease. For instance, the median abundance for *L. lactis* is higher in unaffected children from the low prevalence zone compared to both affected children from the same zone or unaffected children from the high prevalence zone. The fact that the authors link results on these *Lactobacillus* and *Lactococcus* taxa and disease is then unsupported.

I am also skeptical about the computational processing of the shotgun metagenomic data. The authors report an average of >4,000 species in the microbiome of individuals, which is an order of magnitude higher than what's usually being found and reported in the literature. Such

overestimations in the number of species likely results from inadequate use of filtering thresholds in the OTU calling pipeline. OTU calling methods that use a reference database (such as Kraken2 used by the authors) are known to report false positive taxonomies owing to wrong read mappings. Did the authors use a threshold to exclude OTUs with less than X number of reads? This can be done with Bracken, for instance. As an illustration of my criticism, the authors report relative abundances for *Thermodesulfobacterium geofontis* across all individuals in the cohort. But *T. geofontis* is an hyperthermophilic and sulfate-reducing organism, which doesn't occur in the gut. This is just one example of false hits that the authors report and analyzed in their study. I suggest re-running the whole pipeline and all analyses throughout the paper with updated data. This could actually help discovering associations with host traits, such as Konzo status.

Fig 2 to 4 do not present novel findings, and confirm that geography is a major contributor of interindividual differences in microbiome compositions between individuals. The authors should consider presenting these confirmatory data in a more condensed way so the manuscript can put a stronger emphasis on what's new, i.e. the search for associations between the microbiome and Konzo.

As of now, the authors only looked for associations between taxa and traits. But considering that specific metabolic enzymes could make the link between the microbiome and Konzo, such as linamarase, the authors should also search for associations between gene families and traits, not just taxa. This could help discovering more relevant associations in the context of this disease.

L233-241: based on these data, it looks like the population from the ULPZ exhibit classic signatures of adaptation to more urban lifestyles: higher levels of faecalibacterium and lower levels prevotella – can the authors comment on this?

L248: the authors should provide data on diet, nutritional intake and lifestyle to being able to claim that populations from the low and high prevalence regions are homogeneous for all these factors.

L62-64: add a reference. Also, the connection between this sentence and the rest of the section is unclear. Please rephrase for clarification.

L86-87: 'African continent' sounds inappropriate considering that data cover individuals from DRC only.

What does the 'Run' column in the Sample metadata table correspond to?

Legend of Figure 5 is incorrect – there are three panels in the figure, and only two are presented in the legend.

Response to Reviewer Comments:

June 16, 2021

Manuscript ID: NCOMMS-20-35020A

Manuscript Title: The Gut Microbiome in Konzo

Authors: Matthew S. Bramble & Neerja Vashist et al.

Summary of updates to manuscript: Given some overlap of questions from both reviewers 1 and 2, we felt it would be best to initially summarize the major changes to the updated manuscript to better address the points that were raised.

1) In regards to diversity measures and species abundance, in the updated manuscript you will see that as opposed to presenting those data in a figure with lenient filtering (standard output from Bracken), we have now filtered each of the 180 samples to include only species whose abundance is $\geq 0.01\%$. This filtering approach now left ≥ 450 unique species per group, which, as reviewers noted, is far more in line with published findings and is the most appropriate way to analyze such diversity measures, to limit false hits which could skew diversity. While those original presented data yielded ≥ 4000 identified species, downstream analysis on group comparisons was filtered in similar fashion that included only taxonomic ranks that were $\geq 0.01\%$ relative abundance in at least 1 of the 6 study groups, which at the species level only includes 694 bacteria. However, as reviewers noted, these filtering and trimming details were not overly articulated in the methodology of the original manuscript, which has now been adjusted.

2) In the original presentation, when conducting comparisons between groups, we corrected all pair-wise statistics using Benjamini-Hochberg (BH) with a false discovery rate correction set to 0.05. However, in the updated manuscript we wanted to be more statistically conservative, and have since adjusted the FDR from 0.05 to 0.01 for statistical cutoffs, which significantly reduced the number of differences between groups in some instances, and removed focus from particular aspects of the work such as the findings regarding *Turicibacter*. As this is the first investigation into the gut microbiome of children in the DRC as well as those with konzo, we felt that it was best to focus the work on the most statistically sound differences in an effort to highlight the strongest bacterial variations between our study groups.

3) Both reviewers noted the importance of the inclusion of specific gene/functional differences, to add additional insight into how the microbiome may serve to modulate konzo. In the revised manuscript you will now see the inclusion of such data, which revealed very interesting findings. We see that in the Kahemba region (both high and low prevalence zones) as a whole and regardless of disease status, those children have a significantly higher abundance of linamarase (β -glucosidase) representation. This furthers the notion that the microbiome may indeed make those individuals in the Kahemba region more susceptible to developing konzo, given their gut microbiomes have a potentially higher capacity to lyse linamarin in the gut and subsequently release larger proportions of cyanide. Interestingly, we also observed that the children of Masi-manimba, whose diet and lifestyle closely resembles those children of Kahemba, yet outbreaks of konzo are not documented, have significantly increased abundance of Rhodanese and 3-mercaptopyruvate sulfurtransferase (MPST). These two enzymes are key to cyanide detoxification and may shed light as to why the children of Masi-manimba do not have outbreaks of konzo, despite similar diets and lifestyle practices. Collectively, the addition of the metagenomic findings coupled with bacterial abundance differences raise two interesting

hypotheses and scenarios as to how the gut microbiome may serve as a modulator of konzo susceptibility through both protective and susceptibility mechanisms (see discussion in revised manuscript). Below are more detailed and focused responses to other specific concerns raised by both reviewers, where references to line numbers correspond to document with tracked changes.

Reviewer #1:

In this manuscript the authors describe the microbiome of Africans in DRC from an urban, rural, and two villages with a high incidence of Konzo. They find a difference in microbiota composition that is mostly dependent on geography and urbanization and very little difference in microbiota composition between individuals with Konzo and those as yet diagnosed.

1.1: It is difficult to assess many of the conclusions given the lack of details provided in the methods and the somewhat outdated analysis pipeline used.

Response 1.1: We thank the reviewer for this comment and would like to note the importance of understanding the yet diagnosed component, as this is an important aspect of the work, given that the controls for this study in the Kahemba region are certainly still at risk of developing konzo. Our findings indicate high similarity between Kahemba controls and those with konzo, which raises even more concerns as to the susceptibility of developing konzo for those yet diagnosed, if the microbiome is a modulator of disease as the data indicate. We apologize for the brevity of the methods in the original submission and feel that most issues raised were due in part to our lack of methodological detail, rather than an incorrect informatic approach. These issues have now been addressed in the methods section and hopefully have clarified some of the confusion raised by both reviewers. Regarding the major pipelines used, Kraken 2 and Bracken applications perform quite well for metagenomic assignments (Wood et.al and Lu et.al) and we think some of the “pipeline” confusion likely stemmed from the fact this was a metagenomic shotgun approach rather than 16S sequencing as you will see below in response to comment 1.2.

-Wood, D.E., Lu, J. & Langmead, B. Improved metagenomic analysis with Kraken 2. *Genome Biol* **20**, 257 (2019). <https://doi.org/10.1186/s13059-019-1891-0>

-Lu J, Breitwieser FP, Thielen P, Salzberg SL. 2017. Bracken: estimating species abundance in metagenomics data. *PeerJ Computer Science* 3:e104 <https://doi.org/10.7717/peerj-cs.104>

1.2: Specifically, alpha diversity comparisons are highly sensitive to sequencing depth. The authors do not make clear whether any rarefaction took place in order to compare the α -diversity of the different groups. Furthermore, it does not appear that any trimming of low-abundance reads occurred for the α -diversity calculations. Given that the authors use OTUs in lieu of the more commonly accepted DADA2 for “species” it is highly likely that many of the OTUs are in fact spurious and may not represent true diversity. See Prodan et al, (2020) Comparing bioinformatic pipelines for microbial 16S rRNA amplicon sequencing and Callahan et al, (2017) Exact sequence variants should replace operational taxonomic units in marker-gene data analysis for why OTUs are problematic.

Response 1.2: We thank the reviewer for this comment and understand why it was raised in our original submission. In the original presentation of alpha diversity and identified species, the labeling should not have been OTUs but rather “species”, as this is a metagenomic sequencing approach and not 16S sequencing, so this error has now been corrected. In the original submission, the findings were presented using standard trimming from Bracken, which resulted in >4000 species, however as the reviewers note, many of these were likely false hits, which we agree with. We agree that while we filtered data prior to comparative analysis it would also have been best to filter upstream of alpha diversity analysis and have now done so in the revised manuscript. As noted in the above summary of changes, for each of the 180 samples, we filtered taxonomies to include bacteria that were $\geq 0.01\%$ relative abundance prior to performing diversity calculations. This approach resulted in >450 unique species identified per group, as opposed to the original 4000 that was originally presented. This now reflects a more realistic microbiome composition and downstream conclusions. We agree with the reviewer that if these had indeed been OTUs, the suggested approach of DADA2 and exact sequence variants would have been appropriate. However, given that metagenomic sequencing was performed, this is not the case here. In the revised submission we used a relative abundance approach for analysis and not rarefaction, as rarefaction can be equally problematic (McMurdie et.al 2014, Willis 2019), particularly in this case of shotgun metagenomic sequencing where reads are not uniform/focused on the V regions of the genome. In summary, we have now clarified and corrected this component of the work in the revised manuscript (Results, Lines 128-134, Methods, Lines 720-724, Figure 2A/2B). However, regardless of filtering, the final conclusions on diversity and group differences remained the same, speaking to the performance of diversity functions.

- McMurdie PJ, Holmes S (2014) Waste Not, Want Not: Why Rarefying Microbiome Data Is Inadmissible. *PLoS Comput Biol* 10(4): e1003531. <https://doi.org/10.1371/journal.pcbi.1003531>

- Willis AD (2019) Rarefaction, Alpha Diversity, and Statistics. *Front. Microbiol.* 10:2407. doi: 10.3389/fmicb.2019.02407

Updated results lines 128-134: “After filtering to include bacterial taxonomic assignments that were present to at least 0.01% relative abundance in each individual, we observe that all study groups regardless of living environments harbored on average over 450 unique bacterial species (Fig. 2A) (Supplementary File 2). All study groups also displayed measures of α -diversity as measured by the Shannon index that were indicative of a diverse microbial ecosystem (Fig. 2B).

Updated methods lines 720-724: “to better estimate species richness in the sample and remove likely superfluous low abundance taxa, all species with a relative abundance less than 0.01% for a sample were set to zero, only for calculating number of species and the Shannon Diversity Index”.

1.3: Presenting data as shown in Figures 2C-E is problematic since these graphs do not reveal associated error (nor is it mentioned in the text). Differences in relative abundance of phyla, families, and genera mentioned in the manuscript are difficult to interpret if the authors don't reveal the error associated with these percentages. The difference in Bacteroides/Prevotella ratio in the Supplement is the most convincing given the statistically analysis associated with those data and that the finding is in line with that reported now in several manuscripts across not only African populations but those in South America and Asia.

Response 1.3: We agree with the reviewer that there is need to show the error/deviation associated with each level of taxonomy. However, given the amount of data presented, showing errors for each of these in a stacked bar format is quite cluttered as well as difficult to address each component of taxonomy in text, but of course needed. As such, we have included all measures of standard deviation that is associated with taxonomic abundance measures in supplementary file 2, from phylum to species for each study group. This is indeed a relevant point of data, as the microbiome is highly diverse from person to person, so understanding how different taxonomies varied as a group is of interest and is now included in the revised work. We also agree that the B/P ratio data is interesting as well, as it adds to the growing body of literature demonstrating such urban and rural differences even in populations that are understudied as is the case in DR Congo, making it one of the more consistent findings in gut microbiome studies to date.

1.4: The number of genera that are reported to be different between groups is quite high (data associated with Fig 3) and likely a result of spurious OTUs. This analysis should be redone with ASVs to have a more reliable count of ASVs thereby making comparisons between groups more robust. It is also important to note that many differences described are from very low abundance taxa therefore it is incredibly important that care is taken to make sure these are in fact reliable “species”.

Response 1.4: We appreciate this comment from the reviewer and note that while some of the original findings likely contain spurious species as do all studies, those differences presented were based on post trimming ($\geq 0.01\%$ in 1 of 6 groups). As noted in the above summary, we opted for this revision to be more statistically conservative and now corrected all comparisons when using the MWW test with a BH and FDR of .01 as opposed to .05, which significantly reduced the number of differences at this statistical cut off. However, even with more stringent statistical cutoffs, the differences are still large in some cases (>150 genera differences), so this is likely demonstrating that large and true differences exist between some of the study group comparisons in this work. Again, these are not OTUs from 16s sequencing, but rather species post-metagenomic sequencing, so ASVs would not be appropriate in this case, but of course they would be if this was microbial V region sequencing.

1.5: The authors conclusions from Fig7 are problematic in the absence of metagenomic data. The authors are assuming that sequenced strains from western equivalent taxa are similar enough to extrapolate the genomic content from strains coming from individuals from DRC. There is in fact evidence from comparisons of genomes from strains of *Prevotella* from western and non-western samples that this is not necessarily the case. See Tett et al., (2019) The *Prevotella copri* Complex Comprises Four Distinct Clades Underrepresented in Westernized Populations.

Response 1.5: We thank the reviewer for this important comment. We agree that there indeed could be strain differences as noted in the referenced *Prevotella copri* studies, however, our general conclusion is that the observed increased abundance in the lactic acid bacteria (LABs) are generated from cassava consumption and the referenced studies are based on works from Africa and not western strains. While there could be strain differences, we have now included the metagenomic profiles of key genes in the linamarin – cyanide pathway in figure 6. Based on these updated data we see that those individuals in both konzo prevalence zones of Kahemba, regardless of disease status, have a significantly more abundant β -glucosidase (linamarase) representation than children from Masi-manimba, the best control for this study. We interestingly also see increased Rhodanese & MPST gene abundance in the microbiomes of children of Masi-manimba, the key enzyme needed for cyanide detoxification, perhaps contributing to the

lack of konzo in that region of the DRC. Collectively, as the reviewer noted, the addition of the functional component of the metagenomic data greatly strengthened the original hypotheses. It now becomes less important as to whether the LAB's are the contributor of increased linamarase abundance considering the additional data demonstrating higher abundance of the key enzyme needed for cyanide liberation in the gut. This ability to lyse cyanogenic glucosides is likely a process that many bacteria can possess, yet exact details of this process have mostly been studied in Lactic acid bacteria, therefore identifying additional species that effectively lyse linamarin is a great future study direction, which could shed additional light on the influence of gut bacteria in relation to konzo.

1.6: First, the statement: "We showed that regardless of region, all study groups on average harbor >4000 mapped OTUs with high levels of α -diversity, indicative of "healthy" microbiomes." Has several issues. Something can have a high level of diversity by comparison to something that is lower, however there is no comparison in this study. Given the sensitivity of α -diversity measures to read depth and how species are determined in this study, reporting a number for species and saying it is "high" is not appropriate. (Authors also refer to "high" alpha diversity within the main text at Line 121.) Second, the authors need to be very careful in equating "high" diversity and good health in a population for which this has not been studied. Associations between microbiota diversity and health have been exclusively performed in western populations. Given the significant difference in the microbiome of more traditional populations from that of westerners, one cannot assume that high diversity in the traditional population microbiota is also associated with better health.

Response 1.6: We fully agree with this concern and have adjusted the wording of this section to be more reflective of the reality, in that based on diversity measures these individuals as a whole have "diverse" microbial ecosystems (Results lines 132-134). As this work is not necessarily focused on differences in global alpha diversity, which as the reviewer notes is subject to many variables such as sequencing depth etc., we feel that simply stating the outcome of the data is far more advantageous than alluding to high or low diversity type statements. We also fully agree that we cannot know in the current state of microbiome research how high diversity equates to health in understudied populations, because even in our data, we see diverse bacterial profiles in the children of Kahemba, but we know that the children who reside in Kahemba regardless of disease status are by no means "healthy", adding to the reviewers point.

1.7: The section entitled: "Gut bacteria capable of hydrolyzing linamarin are enriched in the Kahemba region" proposes that enrichment of organisms within the microbiota known to harbor linamarases could release additional cyanide within the gut. This hypothesis is extremely premature given the data presented. First, it assumes that strains they detected also carry linamarases. The studies they reference are from strains isolated from cassava not the human gut. Metagenomic sequencing or strain isolation and whole genome sequencing would be required to address whether there is in fact increase linamarases within the microbiome of these participants or whether the strains of *Lactobaccillus*, *Lactococcus*, or *Leuconostoc* harbored within their microbiota do in fact have linamarase activity.

Second, it assumes that linamarase activity within the gut is a major source of cyanide release.

Response 1.7: We thank the reviewer for this comment, and the assumption that bacterial linamarase activity is a major source of cyanide release is based on studies on various livestock ruminal fluids and intestinal scrapings in the 1970/80's. These and other works demonstrated

the ability of cyanide liberation in the guts of animals from bacterial components, as well as numerous rodent studies (below is a recently published comprehensive review on this topic covering the above-mentioned literature, Cressey et.al). Inherently, humans are incapable of lysing the bond in linamarin to release cyanide, and in the absence of a microbiome this cyanogenic glucoside would simply be excreted through urine. However, gut bacteria are indeed capable of breaking the bond of linamarin and liberating cyanide in the guts of mammals and humans are likely to be no different. These previous studies should have been better put into the context of this original work to demonstrate that while it has yet to be studied in a human model, due to the fact that few humans consume toxic cassava as their sole food source, the idea of the involvement of gut bacteria in this process is long-standing, yet has not been frequently studied in recent years. We have now amended the discussion to better reflect these points and to put into context prior studies that led to some of the hypothesis of this manuscript. Given the updated data regarding the functional component of the metagenomic sequencing, we observe trends in gene abundance that may modulate konzo in either protective or susceptibility scenarios, making the exact species that contribute to these findings less important. Our leading hypothesis is that these representations of Lactic acid bacteria may be acquired from the monotonous consumption of cassava which typically undergoes fermentation processes. While all the referenced studies have performed biochemical analysis on strains isolated from cassava, our assumption is that the elevated levels in these children are from cassava consumption, making the species in question likely the same. However, we agree that isolation and whole genome sequencing of these specific strains would indeed be useful but given the sample preparation methods in limited resource settings, it makes the recovery and isolation of these exact species very difficult (and not possible here). However, considering the results of the functional gene analysis, the exact contributors of increased linamarase abundance become less important as previously noted, considering the likelihood that many species (yet to be biochemically studied) harbor this ability to lyse linamarin as well. Identifying additional species that harbor these capacities in the human gut microbiome and studying them in relation to a konzo context will make for exciting future studies and possibly identify major bacterial populations that may modulate konzo susceptibility.

- Cressey P, Reeve J. Metabolism of cyanogenic glycosides: A review. *Food Chem Toxicol.* 2019 Mar;125:225-232. doi: 10.1016/j.fct.2019.01.002. Epub 2019 Jan 4. PMID: 30615957.

- Michlmayr H, Kneifel W. β -Glucosidase activities of lactic acid bacteria: mechanisms, impact on fermented food and human health. *FEMS Microbiol Lett.* 2014 Mar;352(1):1-10. doi: 10.1111/1574-6968.12348. Epub 2013 Dec 16. PMID: 24330034.

1.8: Linamarase activity could be measured within the stool samples to determine whether there is in fact elevated activity in those with Konzo.

Response 1.8: We thank the reviewer for this suggestion. While in theory this is an ideal study, this would require a different type of sample preparation in the field to preserve bacterial and protein function for downstream biochemical measures, which is difficult in resource-limited settings such as Kahemba, and it was not done for this study. Had we known what the data would show in our analysis, this would have been an approach most interesting to pursue. However, this is indeed a measurement that we are planning for future studies when we return to Kahemba for follow up research. We do appreciate the thoughts of the reviewer for this work and suggestions for research strategies that will greatly help to bring clarity to some of the mechanisms at play in the development of konzo.

1.9: Measuring consumed amounts of linamarin would be important to rule out the simplest explanation, which is that those with Konzo are consuming more linamarin in general, perhaps from less fermented cassava. It is also possible that those with Konzo are lacking protective strains (assuming the microbiota is involved in onset or severity of this disease). In other words, the data presented here does not provide a compelling hypothesis to explain a microbiota link to Konzo.

Response 1.9: We thank the reviewer for this comment. Many prior epidemiological studies predominately from one of the corresponding authors, Dr. Desire Tshala, have addressed this point in depth, and are referenced throughout the manuscript (Kashala-Abotnes et.al, Kassa et.al, Boivin et.al). He and his team have demonstrated that the amounts of thiocyanate (indirect measure of linamarin consumption) in the urine of children from Kahemba are significantly increased compared to other populations in the DRC, furthering the association with cassava and konzo. He has also demonstrated that if cassava flour undergoes a secondary detoxification method, known as the wetting method, then reductions in urinary thiocyanate concentrations can be achieved, demonstrating that toxicity content in cassava is the main source of cyanide intoxication in these populations (Banea et.al). Studies have also measured toxin content in cassava flour for the homes of residents of Kahemba and demonstrated the cassava type and preparation leaves considerable amounts of linamarin and cyanohydrin behind, leading to toxicity. The true interest lies in fact that the region consumes similar types of toxic cassava with similar preparation methods, yet 5-10% percent of individuals get konzo during outbreaks while others don't; the reasons behind this remain unknown. The reviewer highlights another very interesting point that perhaps the children of Kahemba are lacking protective bacterial strains. What these protective strains are remains to be seen, but with the addition of functional gene analysis, we see that compared to Masi-manimba, the children of Kahemba have significantly lower abundance of key genes needed to detoxify cyanide such as bacterial Rhodanese and MPST (Results, lines 387-404, Figure. 6B/C). It is very possible that the children of Kahemba have a two-factor problem, in that they have reduced representation of such "protective" strains capable of increased detoxification, while harboring bacterial strains in higher abundance that can lyse linamarin effectively. Collectively the suggestions raised by the reviewer are exactly the types of scenarios that would modulate konzo and with the additional of functional gene data coupled with specific bacterial abundance differences, we feel that the case for a bacterial component in relation to this multifactorial disease is now far more compelling.

-Kashala-Abotnes E, Okitundu D, Mumba D, Boivin MJ, Tylleskär T, Tshala-Katumbay D. Konzo: a distinct neurological disease associated with food (cassava) cyanogenic poisoning. *Brain Res Bull.* 2019;145:87-91. doi:10.1016/j.brainresbull.2018.07.001

-Kassa RM, Kasensa NL, Monterroso VH, Kayton RJ, Klimek JE, David LL, Lunganza KR, Kayembe KT, Bentivoglio M, Juliano SL, Tshala-Katumbay DD. On the biomarkers and mechanisms of konzo, a distinct upper motor neuron disease associated with food (cassava) cyanogenic exposure. *Food Chem Toxicol.* 2011 Mar;49(3):571-8. doi: 10.1016/j.fct.2010.05.080. Epub 2010 Jun 9. PMID: 20538033; PMCID: PMC2962701

-Boivin MJ, Okitundu D, Makila-Mabe B, Sombo MT, Mumba D, Sikorskii A, Mayambu B, Tshala-Katumbay D. Cognitive and motor performance in Congolese children with konzo during 4 years of follow-up: a longitudinal analysis. *Lancet Glob Health.* 2017 Sep;5(9):e936-e947. doi: 10.1016/S2214-109X(17)30267-X. PMID: 28807191; PMCID: PMC5594926.

-Banea JP, Bradbury JH, Mandombi C, Nahimana D, Denton IC, Kuwa N, Tshala Katumbay D. Control of konzo by detoxification of cassava flour in three villages in the Democratic Republic of Congo. *Food Chem Toxicol.* 2013 Oct;60:506-13. doi: 10.1016/j.fct.2013.08.012. Epub 2013 Aug 11. PMID: 23941775.

Updated results lines 387-404: “Given that, we next sought to determine if sequences that mapped to β -D-glucosidase (EC: 3.2.1.21) (KO 5350) genes were also enriched in Kahemba. Interestingly, we observe that when compared to Masi-Manimba, a village whose diet most closely resembles that of Kahemba, genes that code for β -D-glucosidase (EC: 3.2.1.21) are enriched in unaffected and konzo-affected children from both the LPZ (BH- Corrected MWW $p=0.013$, $p=0.028$, respectively) and HPZ (BH-Corrected MWW $p=0.034$, $p=0.078$, respectively) (Fig. 6B) (Supplementary File 5). While some bacteria harbor the potential to exacerbate the effects of linamarin exposure by harboring β -D-glucosidase enzymes, other bacteria have been shown to harbor the ability to detoxify cyanogenic compounds via pathways utilizing thiosulfate sulfurtransferase/Rhodanese (EC: 2.8.1.1) and 3-mercaptopyruvate sulfurtransferase/MPST (EC: 2.8.2.1). When compared to the unaffected and konzo affected children residing in the LPZ (MWW $p=0.007$, $p=0.016$, respectively) and HPZ (MWW $p=0.008$, $p=0.002$, respectively) of Kahemba, the children of Masi-Manimba on average have significantly more abundant representation of both bacterial MPST and Rhodanese genes (KO 1011) (Fig. 6C) (Supplementary File 5). Collectively, these data highlight two plausible scenarios as to how the gut microbiome can modulate the development of konzo, through either a susceptibility or protective scenario, under the assumption that all other required factors are present that enable the development of konzo”.

Minor points:

1.10: What is the y-axis in Supplementary Figure 1? Servings? How is this normalized?

Response 1.10: The y-axis represents number of days per week that a particular food was consumed, so there is not any normalization per se, but rather just an average amount of days each type of food was consumed per week for each of the study groups. The major point of this figure was to highlight the lack of food diversity in the Kahemba region, particularly the lack of foods rich in sulfur containing amino acids such as meat and dairy, a key component needed in the detoxification process of cyanide.

1.11: The colors in Fig 3 make it very hard to discern groups.

Response 1.11: We have now attempted to make the colors more distinguishable, but if difficulty remains, the color scheme can be overhauled.

1.12: Figure 5 legend is confusing and doesn't refer to all panels.

Response 1.12: Agreed and has now been corrected in the updated manuscript.

1.13: Fig 7B is significant relative to Kin or Mas or both? For all 3 strains graphed? Need more clarity for these data.

Response 1.13: Figure 7 (now figure 6) has undergone significant changes and should now be clear on what groups are being compared to one another.

1.14: Line 311: Authors report that *Leuconostoc mesenteroides* is more abundant in children from Masi-Manimba as compared to Kinshasa. Is there is difference in Konzo prevalence between these 2 areas?

Response 1.14: The p value = 0.02 for the difference between these two sites, so we still consider children in Masi-manimba to have significantly more abundant *Leuconostoc* when compared to Kinshasa. However, there are no documented outbreaks of konzo in either

Kinshasa or Masi, but there are instances of sporadic cases in both, as poverty is a driving force behind konzo, forcing individuals to consume exclusively cassava as their only food source.

Reviewer # 2

In this paper, Matthew Bramble, Neerja Vashist and colleagues investigate the association between the gut microbiome and Konzo, a neurodegenerative disease mostly affecting children and caused by the ingestion of food toxins. Konzo mostly occurs in low-income regions, and the authors sampled cohorts in different urban and rural regions in DRC, including from areas with no, low and high prevalence of Konzo.

2.1: The main strength of the paper is to present microbiome data from populations that are seriously under-characterized in microbiome science across a range of different factors: lifestyle (non-industrialized lifestyle), geography (DRC, and urban vs. rural), age (this is an infant cohort) and disease status (Konzo is under-studied, especially on its connection to the gut microbiome). The design of the cohort is thoughtful: the authors recruited participants in both urban and rural regions and in both low and high prevalence regions. In each of these sampling areas, a similar number of participants (30) have been recruited. This recruitment scheme should open the possibility of teasing apart contributions of lifestyle, geography and disease status on inter-individual differences in the microbiome, which could make the study quite powerful.

Response 2.1: We thank the reviewer for the positive comments and fully agree that the data set has the potential to uncover inter-individual components. However, given the number of uncontrolled variables in human microbiome studies, it makes teasing out the exact contributors difficult. Additionally, while outside the scope of the initial study the intra-individual differences observed in each group of study are also quite interesting outside of the context of konzo, as we see high levels of bacterial similarity in rural areas compared to heterogeneity in urban study groups.

2.2: However, I'm not convinced that the authors effectively disentangled the contribution of geography (and/or diet) from the contribution of disease on differences in microbiome compositions. No results currently presented in this paper support a potential association between the microbiome and Konzo. I am also not convinced that the authors have the necessary data types to discover such associations, if there are any.

Response 2.2: We thank the reviewer for this comment. We agree that, it is virtually impossible to disentangle geography from diet, as individuals living in different locations rely on cassava with different levels of toxicity, so diet & geography are essentially one singular factor which has been treated as such. The goal of this work was not to ask if the microbiome was the cause of konzo, but rather a likely modulator. Additional evidence in the revised manuscript using functional gene assessment from the metagenomic data revealed that the children of both the high and low prevalence zones of Kahemba show increased representation of the key enzyme needed for the lysis of linamarin (β -glucosidase), a possible mechanism for their higher susceptibility to konzo. Alternatively, we also observe that the children of Masi-manimba who consume similar cassava, with yet no documented konzo outbreaks, have significantly higher representation of bacterial rhodanese, the key enzyme needed for detoxification. As to whether these genes are truly functional will certainly require additional biochemical experimentation, but as of now, certainly raises a plausible hypothesis. The idea of the involvement of the gut bacteria in relation to cyanide release stems from the 80's, yet this is essentially the first assessment in humans, as was described in more depth in responses to reviewer 1. There is

ample evidence that gut bacteria are responsible for cyanide release in the guts of humans, as humans do not inherently possess the ability to lyse linamarin, however its exact contribution to konzo development will require more in-depth biochemical investigations. This is further complicated by the multifactorial nature of konzo, where the microbiome plays a role, along with the additional factors such as malnutrition, sulfur amino acid deficiency and consuming improperly processed cassava, making the disentanglement of these key causative agents quite difficult.

2.3: Overall, the authors should use statistical techniques to account for multiple factors at once and measure the effect size and p-value of each individual factor. Right now, the authors performed analyses separately for each factor, which prevents the accurate measurement of the real contribution of each variable, especially disease status. Factors should include disease status, geographic region of sampling, age, sex, diet, etc.

Response 2.3: We thank the reviewer for this comment and agree. In the original manuscript we did not explicitly explain why we compared groups the way we did. In the revised manuscript we do now note this key point, to better clarify as to why analysis was done the way it is (Results, lines 169-171). When testing comparisons to include all possible interactors, it appears that no other factor has a significant co-interaction aside from geography/diet (see results from test below), which is the reason we measure this component in all of our comparisons. If these statistical tests are of interest, the specific results can be provided, however all available data will be freely accessible should other researchers want to test these comparisons as well.

```
> otuD.G <- as.data.frame(t(otu_table(KonzoData.G)))
> diversity.G <- estimate_richness(KonzoData.G)
> diversity.G <- cbind(sample_data(KonzoData.G),diversity.G) #Might change since cbind can be tricky and not reliable, so always confirm
if correctly done
> brayd <- phyloseq::distance(KonzoData.G.tr, method="bray")
> bdiv_bray <- adonis(brayd ~ diversity.G$Geography * diversity.G$Region * diversity.G$Disease * diversity.G$Age * diversity.G$Sex, perm=
99999); bdiv_bray
```

```
Call:
adonis(formula = brayd ~ diversity.G$Geography * diversity.G$Region * diversity.G$Disease * diversity.G$Age * diversity.G$Sex,
permutations = 99999)
```

```
Permutation: free
Number of permutations: 99999
```

```
Terms added sequentially (first to last)
```

	Df	SumsOfSqs	MeanSqs	F.Model	R2	Pr(>F)
diversity.G\$Geography	3	1.6130	0.53767	6.7170	0.10444	1e-05 ***
diversity.G\$Disease	1	0.0498	0.04982	0.6224	0.00323	0.7325
diversity.G\$Age	1	0.0581	0.05809	0.7257	0.00376	0.6362
diversity.G\$Sex	1	0.0992	0.09925	1.2399	0.00643	0.2605
diversity.G\$Geography:diversity.G\$Disease	1	0.0323	0.03230	0.4035	0.00209	0.9160
diversity.G\$Geography:diversity.G\$Age	3	0.1670	0.05567	0.6955	0.01081	0.8272
diversity.G\$Disease:diversity.G\$Age	1	0.0579	0.05792	0.7235	0.00375	0.6361
diversity.G\$Geography:diversity.G\$Sex	3	0.1819	0.06064	0.7576	0.01178	0.7595
diversity.G\$Disease:diversity.G\$Sex	1	0.0489	0.04886	0.6104	0.00316	0.7444
diversity.G\$Age:diversity.G\$Sex	1	0.0629	0.06286	0.7853	0.00407	0.5793
diversity.G\$Geography:diversity.G\$Disease:diversity.G\$Age	1	0.0493	0.04935	0.6165	0.00320	0.7358
diversity.G\$Geography:diversity.G\$Disease:diversity.G\$Sex	1	0.0569	0.05692	0.7111	0.00369	0.6473
diversity.G\$Geography:diversity.G\$Age:diversity.G\$Sex	3	0.3302	0.11007	1.3751	0.02138	0.1305
diversity.G\$Disease:diversity.G\$Age:diversity.G\$Sex	1	0.0912	0.09123	1.1397	0.00591	0.3145
diversity.G\$Geography:diversity.G\$Disease:diversity.G\$Age:diversity.G\$Sex	1	0.0584	0.05840	0.7295	0.00378	0.6271
Residuals	156	12.4873	0.08005		0.80853	
Total	179	15.4444			1.00000	

```
---
Signif. codes: 0 '***' 0.001 '**' 0.01 '*' 0.05 '.' 0.1 ' ' 1
```

Results lines 169-171: “After accounting for all possible interactors such as age, sex, location and disease status, our data indicate that geographic location (cassava toxicity) is the variable that significantly contributes to observed bacterial composition differences”.

2.4: Section L262-275 and figure 5/6: the comparison between all unaffected and all affected children is lacking. This is the control experiment that we need to see whether individuals with disease state have higher microbiome similarity independently of geography of origin. Had the authors analyzed their data with models that handle the simultaneous effect of multiple variables, the measurement of the effect size of geography vs. diet vs. disease would have been made possible.

Response 2.4: We thank the reviewer for highlighting this important comparison. As noted, when all factors such as age, sex, geography/diet and disease status are taken into account, geography/diet are the driving force behind observed statistical differences. However, when one compares the unaffected children from the High Prevalence Zone (HPZ) to the unaffected children from the Low Prevalence Zone (LPZ), their microbial profiles show significant segregation ($p < 0.001$), indicating general bacterial abundance differences between these two Kahemba regions (Results lines: 263-269, Figure 5A). When the same comparison is made for those affected with konzo, we still observe significant segregation but to a substantially smaller degree ($p = 0.01$), using Bray-Curtis measures (Fig 5C). If unaffected groups and affected groups are put together and compared to either Kinshasa and Masi-manimba, significant segregation remains, likely due to geographic/dietary differences. In the revised manuscript however, using more conservative statistical cut offs for bacterial inclusion, we observe in pair-wise assessments 69 genera that remain statistically different between unaffected children from the LPZ versus the HPZ. Interestingly, we now observe zero statistical differences in genera abundance for those children with konzo from the LPZ and HPZ. Collectively, just based on this level of comparison, it appears as suggested by the reviewer that children with konzo have a more similar gut microbiome than children who are unaffected. Why this is remains unknown, but could be a post-konzo effect, as mobility is reduced in these children, social stabilization becomes present, along with an array of other variables to consider when being handicap in such a poverty-stricken region of the Congo. Future studies will attempt to determine if similarities in the microbiome are inherent prior to the onset of the disease, or if having the disease makes individuals more similar, both very interesting points of study.

2.5: Another example of inappropriate dissection of the factors: Fig 7A does not show whether the relative abundance of Turicibacter is significantly higher in affected children of the low prevalence zone (KLPZ) compared to unaffected children of the high prevalence zone (UHPZ). This is necessary to exclude an effect of village/geography.

Response 2.5: We agree, however, in the revised manuscript we decided to be more conservative in our statistical cutoffs and used a post-MWW BH correction with an FDR = 0.01 as opposed to 0.05. When we do this, Turicibacter no longer remains significant and we have since dropped this particular line of investigation and corresponding graphic representation, as we feel it is best to focus on the strongest statistical differences and the newly updated functional gene differences, as opposed to species that likely have little to no role in the modulation of konzo.

2.6: Fig. 7B is also another illustration of a strong effect of geography, and not disease. Many comparisons don't go the way expected if there was an effect of disease. For instance, the median abundance for L lactis is higher in unaffected children from the low prevalence zone compared to both affected children from the same zone or unaffected children from the high prevalence zone. The fact that the authors link results on these Lactobacillus and Lactococcus taxa and disease is then unsupported.

Response 2.6: We thank the reviewer for this comment. Overall, this is indeed what we observe in the data. However, a key point is that the unaffected children in both regions of Kahemba are in theory at equal risk for developing konzo, as their diets and lifestyles are nearly identical to the children that already have konzo. While no single difference in bacteria species is likely to cause konzo, the fact that we see these potentially detrimental bacteria highly abundant in unaffected children from either zone, is cause for concern. If these and other strains of bacteria are major contributors of cyanide release within the guts as we and others hypothesize, then the unaffected children who have heightened abundance of such are at most risk when an environmental stressor sets the stage for a konzo outbreak, such as a drought. Given the available data, it is equally possible that at the time when those with konzo were initially affected, they too had even higher levels of such bacteria species. The key point is that regardless of LPZ or HPZ, it appears that all children in the Kahemba region have significantly higher abundances of LABs as well as genes that code for linamarase as compared to other rural villages such as Masi-manimba. Konzo is highly multifactorial, so it is very possible that one of the additional factors that lead to konzo was not present at the time of collection, but should it arise, those with elevated potential to lyse linamarin are likely most at risk. We again would like to stress that the goal of this work is to not claim that specific bacteria directly cause konzo, but rather based on these data and prior literature on the topic, it appears that the gut microbiome has the capacity to exacerbate or reduce the effects of linamarin ingestion. It is also possible that these bacterial abundances are transitory depending on the fermentation levels of the cassava being consumed, so a continual high level of LABs is not necessarily always to be expected unless there is continued consumption of such foods rich in these bacterial species, much like a pro-biotic effect.

2.7: I am also skeptical about the computational processing of the shotgun metagenomic data. The authors report an average of >4,000 species in the microbiome of individuals, which is an order of magnitude higher than what's usually being found and reported in the literature. Such overestimations in the number of species likely results from inadequate use of filtering thresholds in the OTU calling pipeline. OTU calling methods that use a reference database (such as Kraken2 used by the authors) are known to report false positive taxonomies owing to wrong read mappings. Did the authors use a threshold to exclude OTUs with less than X number of reads? This can be done with Bracken, for instance. As an illustration of my criticism, the authors report relative abundances for *Thermodesulfobacterium geofontis* across all individuals in the cohort. But *T. geofontis* is an hyperthermophilic and sulfate-reducing organism, which doesn't occur in the gut. This is just one example of false hits that the authors report and analyzed in their study. I suggest re-running the whole pipeline and all analyses throughout the paper with updated data. This could actually help discovering associations with host traits, such as Konzo status.

Response 2.7: Thank you. Similar and important comments were also raised by reviewer 1 (1.2) and we have copied our answer below in addition to expanding on additional concerns raised here.

Response 1.2: We thank the reviewer for this comment and understand why it was raised in our original submission. In the original presentation of alpha diversity and identified species, the labeling should not have been OTUs but rather "species", as this is a metagenomic sequencing approach and not 16S sequencing, so this error has now been corrected. In the original submission, the findings were presented using standard trimming from Bracken, which resulted in >4000 species, however as the reviewers note, many of these were likely false hits, which we agree with. We agree that while we filtered data prior to comparative analysis it would also have been best to filter upstream of alpha diversity analysis and have now done so in the

revised manuscript. As noted in the above summary of changes, for each of the 180 samples, we filtered taxonomies to include bacteria that were $\geq 0.01\%$ relative abundance prior to performing diversity calculations. This approach resulted in >450 unique species identified per group, as opposed to the original 4000 that was originally presented. This now reflects a more realistic microbiome composition and downstream conclusions. We agree with the reviewer that if these had indeed been OTUs the suggested approach of DADA2 and exact sequence variants would have been appropriate. However, given that metagenomic sequencing was performed this is not the case here. In the revised submission we used a relative abundance approach for analysis and not rarefaction, as rarefaction can be equally problematic (McMurdie et.al 2014, Willis 2019), particularly in this case of shotgun metagenomic sequencing where reads are not uniform/focused on the V regions of the genome. In summary, we have now clarified and corrected this component of the work in the revised manuscript (Results, Lines 128-134, Methods, Lines 720-724, Figure 2A/2B). However, regardless of filtering, the final conclusions on diversity and group differences remained the same, speaking to the performance of diversity functions.

- McMurdie PJ, Holmes S (2014) Waste Not, Want Not: Why Rarefying Microbiome Data Is Inadmissible. *PLoS Comput Biol* 10(4): e1003531. <https://doi.org/10.1371/journal.pcbi.1003531>

- Willis AD (2019) Rarefaction, Alpha Diversity, and Statistics. *Front. Microbiol.* 10:2407. doi: 10.3389/fmicb.2019.02407

Updated results lines 128-134: "After filtering to include bacterial taxonomic assignments that were present to at least 0.01% relative abundance in each individual, we observe that all study groups regardless of living environments harbored on average over 450 unique bacterial species (Fig. 2A) (Supplementary File 2). All study groups also displayed measures of α -diversity as measured by the Shannon index that were indicative of a diverse microbial ecosystem (Fig. 2B).

Updated methods lines 720-724: "to better estimate species richness in the sample and remove likely superfluous low abundance taxa, all species with a relative abundance less than 0.01% for a sample were set to zero, only for calculating number of species and the Shannon Diversity Index".

Response 2.7: As to the second point of the critic regarding *Thermodesulfobacterium geofontis*, these types of bacteria were filtered out prior to any comparative analysis as they did not reach $\geq 0.01\%$ in any of the 6 groups of study. The supplemental files that were originally uploaded included everything from the Bracken output, for transparency, but we completely understand where the confusion arose, as there was indeed room for methodological expansion. In the resubmission, our supplemental files are now those that underwent our trimming/filtering criteria to limit this confusion for future readers and include only taxonomic assignments that passed our filtering criteria used for all downstream comparative analysis in this study. All raw data and files will be freely accessible upon publication, should others wish to analyze this metagenomic sequencing using different approaches.

2.8: Fig 2 to 4 do not present novel findings and confirm that geography is a major contributor of interindividual differences in microbiome compositions between individuals. The authors should consider presenting these confirmatory data in a more condensed way so the manuscript can put a stronger emphasis on what's new, i.e. the search for associations between the microbiome and Konzo.

Response 2.8: We thank the reviewer for this suggestion. While we have attempted to shorten some of these findings and included them in the supplement, the term geography could be confusing. While the study groups are indeed from different geographical regions, this is also a factor contributing to diet. The geography of Kahemba encourages bitter cassava to be grown due to soil conditions, and while the children in this region are geographically different than Kinshasa, the important point is not the longitude and latitude, but rather the toxicity of cassava in the regions. This is the first metagenomic investigation into children in the DRC and in those with konzo, so we have tried to place emphasis on the association between the microbiome and konzo (as suggested by the reviewer), while still presenting -and shortening- the geography/nutrition aspects which still carries novelty. We have removed some figures such as those regarding *Turicibacter* so that a focus can be given to the more interesting aspects such as the LABs and the newly included findings relating to functional gene differences that we observe.

2.9: As of now, the authors only looked for associations between taxa and traits. But considering that specific metabolic enzymes could make the link between the microbiome and Konzo, such as linamarase, the authors should also search for associations between gene families and traits, not just taxa. This could help discovering more relevant associations in the context of this disease.

Response 2.9: We thank the reviewer for this important comment and insight to the study. In the revised manuscript you will see the expansion of exactly what you are suggesting, such as specific metabolic enzymes that could make the link between the microbiome and konzo. While LABs are known to possess the ability to lyse linamarin, this is unlikely unique to just those species. As such, when we measure abundance of functional genes, particularly β -glucosidase (linamarase), we see that the children of Kahemba regardless of disease status have significantly more representation of such enzymes when compared to the most appropriate controls in Masi-manimba. These additional data now at the functional level add more confidence in the notion that the gut flora of this region of the DRC may be one of the several aspects for developing konzo when all other factors are present for disease outbreaks. Interestingly, in this line of thought, we also observe that the children of Masi-manimba harbor an enrichment of bacterial Rhodanese/MPST (Results lines: 387-404, Figure 6C), two of the key enzymes needed for the detoxification of cyanide post linamarin lysis. So it is also possible that the children of Masi-manimba have reduced frequency of konzo by having lower levels of linamarase and heightened levels of detoxifying genes. Collectively, we feel that by incorporating these findings coupled with the bacterial profiles, the argument for the potential detrimental or susceptibility microbiome profiles have garnered significantly more convincing support. Undoubtedly, additional biochemical work will be necessary for future studies, but this metagenomic assessment certainly sheds light on the possible roles of microbial interactors in the context of konzo in the DRC and perhaps other regions of the world, and enable key enzymes and bacteria to be focused on in future study designs.

Updated results lines 387-404: "Given that, we next sought to determine if sequences that mapped to β -D-glucosidase (EC: 3.2.1.21) (KO 5350) genes were also enriched in Kahemba. Interestingly, we observe that when compared to Masi-Manimba, a village whose diet most closely resembles that of Kahemba, genes that code for β -D-glucosidase (EC: 3.2.1.21) are enriched in unaffected and konzo-affected children from both the LPZ (BH- Corrected MWW $p=0.013$, $p=0.028$, respectively) and HPZ (BH-Corrected MWW $p=0.034$, $p=0.078$, respectively) (Fig. 6B) (Supplementary File 5). While some bacteria harbor the potential to exacerbate the effects of linamarin exposure by harboring β -D-glucosidase enzymes, other bacteria have been shown to harbor the ability to detoxify cyanogenic compounds via pathways utilizing thiosulfate

sulfurtransferase/Rhodanese (EC: 2.8.1.1) and 3-mercaptopyruvate sulfurtransferase/MPST (EC: 2.8.2.1). When compared to the unaffected and konzo affected children residing in the LPZ (MWW $p=0.007$, $p=0.016$, respectively) and HPZ (MWW $p=0.008$, $p=0.002$, respectively) of Kahemba, the children of Masi-Manimba on average have significantly more abundant representation of both bacterial MPST and Rhodanese genes (KO 1011) (Fig. 6C) (Supplementary File 5). Collectively, these data highlight two plausible scenarios as to how the gut microbiome can modulate the development of konzo, through either a susceptibility or protective scenario, under the assumption that all other required factors are present that enable the development of konzo”.

Minor Comments:

2.10: L233-241: based on these data, it looks like the population from the ULPZ exhibit classic signatures of adaptation to more urban lifestyles: higher levels of faecalibacterium and lower levels prevotella – can the authors comment on this?

Response 2.10: We agree with the reviewer on this logic, however unfortunately we cannot offer a well-supported explanation for these findings. The ULPZ is actually even more rural than the HPZ of Kahemba, so if anything, we would have expected an opposite outcome. However, we can be certain that neither the HPZ or LPZ are remotely urbanized. Why these profiles appear the way they do remains unknown.

2.11: L248: the authors should provide data on diet, nutritional intake and lifestyle to being able to claim that populations from the low and high prevalence regions are homogeneous for all these factors.

Response 2.11: We agree, and these data points are now included in supplemental file 1.

2.12: L62-64: add a reference. Also, the connection between this sentence and the rest of the section is unclear. Please rephrase for clarification.

Response 2.12: The statement was vague and unconnected, and we agree, so we have since removed it.

2.13: L86-87: ‘African continent’ sounds inappropriate considering that data cover individuals from DRC only.

Response 2.13: Agreed and corrected.

2.14: What does the ‘Run’ column in the Sample metadata table correspond to?

Response 2.14: This was a residual artifact from raw files on the metafile that has since been removed.

2.15: Legend of Figure 5 is incorrect – there are three panels in the figure, and only two are presented in the legend.

Response 2.15: Thank you for noticing the error and this has since been corrected.

REVIEWER COMMENTS

Reviewer #1 (Remarks to the Author):

The authors have addressed all my concerns and given the data presented, I feel this study will be of interest to the scientific community and represents an important advancement in the understanding of the microbiota from non-western lifestyle individuals. I have no further concerns.

Reviewer #2 (Remarks to the Author):

This is a resubmission of a previously submitted paper on the association between the gut microbiome of children from DRC and Konzo, which is a neurodegenerative disease frequently affecting children in DRC who consume large amounts of food-containing toxins.

The addition of the functional profiling of enzyme families involved in the release of toxins through the degradation of linamarin or involved in the detoxification of cyanide is interesting and increases the interest of the manuscript.

The first round of review highlighted several methodological issues with the analysis of microbiome data. Unfortunately, strong issues remain in this revision, preventing meaningful interpretations of the results presented in Fig3 to 6.

First, the issue of including false-positive taxa called by Kraken+Bracken into diversity analyses still holds in the present manuscript, despite what the authors claim in their response. This can be simply seen by comparing PCoA plots showing differences in beta-diversity between groups (Figure 3 and 5) to the panels presented in the previous submission – they are exactly the same (only colors were changed). This is confirmed by what the authors state in their response and in the Methods: filtering out low abundant taxa (<0.01%) from the data has only been done for alpha diversity measurements. But it should also be done for beta-diversity analyses! Because panels are the same, it means that 100s of taxa wrongly detected by Kraken2, such as the *Thermodesulfobacterium geofontis* that I mentioned in my previous review (an hyperthermophilic and sulfate-reducing organism that does not colonize human guts), are still incorporated in analyses shown in this resubmission. As a consequence, all conclusions drawn in Fig 3-5 cannot be evaluated.

Second, it is now very much recognized in the field that microbiome data are both scarce and compositional, and that appropriate data transformation and normalization techniques must be employed to compare the abundance of individual taxa between groups. In this manuscript, no such technique was employed, and only raw relative abundances were compared, which makes analyses inaccurate. Authors should look at recent tools developed for accounting for the compositionality of microbiome data (e.g. Aldex2, Ancov, etc...). Conclusions on the between-group difference in abundance of *Prevotella*, *Faecalibacterium*, *L. mesenteroides*, *L. plantarum*, etc, cannot be interpreted right now (while being potentially true). I should also note that the Methods seem to imply that abundances lower than 0.01% were set to zero when comparing taxa abundances between groups. If this is true, it further increases the issue of data scarcity.

L106: can you show data supporting that people from these two regions do consume similar amounts of casava?

L196: information is missing – “Test, FDR < XX”

Can the authors comment on potential batch effects in their analysis? Were samples from different regions processed in a single or separate batches (for DNA extraction, library prep, sequencing, etc...).

Response to Reviewer 2:

Reviewer #2 (Remarks to the Author):

This is a resubmission of a previously submitted paper on the association between the gut microbiome of children from DRC and Konzo, which is a neurodegenerative disease frequently affecting children in DRC who consume large amounts of food-containing toxins.

The addition of the functional profiling of enzyme families involved in the release of toxins through the degradation of linamarin or involved in the detoxification of cyanide is interesting and increases the interest of the manuscript.

The first round of review highlighted several methodological issues with the analysis of microbiome data. Unfortunately, strong issues remain in this revision, preventing meaningful interpretations of the results presented in Fig3 to 6.

2.1: First, the issue of including false-positive taxa called by Kraken+Bracken into diversity analyses still holds in the present manuscript, despite what the authors claim in their response. This can be simply seen by comparing PCoA plots showing differences in beta-diversity between groups (Figure 3 and 5) to the panels presented in the previous submission – they are exactly the same (only colors were changed). This is confirmed by what the authors state in their response and in the Methods: filtering out low abundant taxa (<0.01%) from the data has only been done for alpha diversity measurements. But it should also be done for beta-diversity analyses! Because panels are the same, it means that 100s of taxa wrongly detected by Kraken2, such as the *Thermodesulfobacterium geofontis* that I mentioned in my previous review (an hyperthermophilic and sulfate-reducing organism that does not colonize human guts), are still incorporated in analyses shown in this resubmission. As a consequence, all conclusions drawn in Fig 3-5 cannot be evaluated.

Response 2.1: We would like to thank the reviewer for not only a thorough review of the resubmission, but also raising points to clarify in the statistical approaches for microbial analysis. As the reviewer notes, in the resubmission the only difference to all the PCoA plots was a change in color scheme. This was done not to mislead, but simply because reviewer 1 had concerns with the initial color choices. In the corrected manuscript all PCoA plots generated using Bray-Curtis measures now only include genera that passed the filtering criteria. Despite filtering out the bulk of genera, the visual representations are virtually indistinguishable from the plots that incorporated the false-positive taxa, which was the initial reason for choosing unfiltered presentations in the original manuscript. However, this was not unexpected considering the genera that passed the filter criteria constituted >95% or higher of all bacterial representation in the groups of comparison, perhaps speaking to the performance and interpretations of Bray-Curtis indices, in regards to the handling of very low abundant false hits such as the *Thermodesulfobacterium geofontis*, among others. Subtle and minor changes that occurred from filtering include the percent variation measures on the X and Y axis for both Fig 3 & 5, along with the genus that was most associated with axis 2 which is now *Faecalibacterium* as opposed to *Lachnoclostridium*, which is now the 2nd most associated taxa with that particular axis (Fig 3). While the updated figure with filtering is visually very close to the original figure generated without filtering, the data incorporated into all the PCoa Bray-Curtis plots for the new resubmission are void of the false hits without altering any of the fundamental conclusions that were drawn from these analyses. Below are the original figures 3 and 5 and the revised updated figures 3 and 5.

Figure 3 in the original resubmission that included all taxa

Figure 3 in the new resubmission that incorporates only genera that passed the filtering criteria.

Figure 5 in the original resubmission that included all taxa (A, C, D, E)

Figure 5: Updated to include only genera that passed filter Criteria (A, C, D, E)

2.2: Second, it is now very much recognized in the field that microbiome data are both scarce and compositional, and that appropriate data transformation and normalization techniques must be employed to compare the abundance of individual taxa between groups. In this manuscript, no such technique was employed, and only raw relative abundances were compared, which makes analyses inaccurate. Authors should look at recent tools developed for accounting for the compositionality of microbiome data (e.g. Aldex2, Ancov, etc...). Conclusions on the between-group difference in abundance of *Prevotella*, *Faecalibacterium*, *L. mesenteroides*, *L. plantarum*, etc, cannot be interpreted right now (while being potentially true). I should also note that the Methods seem to imply that abundances lower than 0.01% were set to zero when comparing taxa abundances between groups. If this is true, it further increases the issue of data scarcity.

Response 2.2: We thank the reviewer for raising this point of concern, and we do agree that these types of metagenomic bacterial distribution data sets are indeed compositional in nature. While we did not incorporate Aldex2-based statistics in the original submission for pair-wise

comparisons, these types of analysis were done to account for compositionality, as seen in Figure 4 (machine learning). However, since this concern has now been raised, we have adjusted all statistical comparisons in pair-wise group differences to reflect outputs using Aldex2, which uses a CLR transformation approach prior to testing group differences. When incorporating comparisons using Aldex2, the main conclusions of the manuscript remain unchanged and, in most instances, adds statistical confidence to the main hypotheses of the work, as Aldex2 and other normalization methods enable differences in lower abundant bacteria to be better reflected. As you will see, we have now updated figures to reflect both relative abundance for more interpretability, as well as provide median CLR values to reflect statistics based on Aldex2. We have also included an additional supplemental figure to accompany Figure 6, which reflects all statistical comparisons based on non-parametric tests (Aldex2) that highlight the enrichment of linamarase positive bacteria in the Kahemba region as compared to non-Konzo areas such as Masi-Manimba and Kinshasa. After filtering to include those genera that passed our initial filter prior to Bray-Curtis analysis in Figure 5, we see that while *Prevotella* and *Faecalibacterium* remain the most associated bacteria with PC Axis 1 and 2, yet *Prevotella* differences no longer reach statistical significance when compositionality is taken into account using Aldex2. While a minor point in the manuscript, we chose to leave both relative abundance and CLR transformed comparisons for this presentation, perhaps adding to the point of the reviewer in that it is important to consider the compositionality of the data set, particularly in regards to highly abundant taxa. We have also provided an additional supplemental file that includes outputs of Aldex2 comparisons in addition to those based on relative abundance, providing future readers of the work both outputs of statistical comparison. As this work is the first investigation into the relationship of the microbiome and konzo, we appreciate the concerns raised by the reviewer and feel that addressing compositionality in the data in this manner adds additional confidence to the hypothesis raised by this work. Below we have included the updates to main figures as well as the updated supplement to figure 6, reflecting the comparisons based on Aldex2 statistics.

Regarding the second point that was raised by the reviewer, we apologize for the confusion as the methods were not clear in that this did NOT apply to taxa abundances between groups. For group comparisons, taxa whose relative abundance was $\geq 0.01\%$ in any 1 of 6 groups was retained for all comparisons, and they were not artificially set to zero, as that would have indeed been statistically improper.

Figure 5:

Updated to display statistical differences based on post-transformed data using Aldex2

Supplementary Figure 7: Supplementary data to Figure 6, using CLR transformed median differences based on the non-parametric statistical outputs obtained after Aldex2 calculations.

L106: can you show data supporting that people from these two regions do consume similar amounts of casava?

Response: Our data indicate that cassava consumption per week is similar across the populations of these two regions. Data are based on the dietary questionnaire that accompanies the manuscript (Supplementary Figure 1). Of course, exactly quantified (by actual weight) amounts of cassava consumption would be a hard measure to obtain. Similar consumption is also consistent with the fact that cassava is the known staple food of impoverished communities such as Kahemba and Masi-Manimba.

L196: information is missing – “Test, FDR < XX”

Response: Thank you for noticing this error, and it has since been corrected.

Can the authors comment on potential batch effects in their analysis? Were samples from different regions processed in a single or separate batches (for DNA extraction, library prep, sequencing, etc...).

Response: Regarding batch effects, we attempted to control for as many variables as possible when conducting this work, from sample collection through sequencing. All samples were

collected by the same operators under the same cold-chain protocols prior to being shipped to USA together. All samples were also extracted, processed and quantified over the same time course (180 samples), by a single operator using the same extraction kits where reagents were pooled (when appropriate) to accommodate the total sample size prior to extraction. At the GWU genomics core facilities, library preparation was processed for all 180 samples before pooling and splitting between runs to obtain similar read depths and quality for each study group.

REVIEWERS' COMMENTS

Reviewer #2 (Remarks to the Author):

The authors have appropriately addressed my last concerns.

As said in the introduction of my first review, I'm enthusiastic about the paper as it addresses a serious medical concern that affects under-represented populations in microbiome research. The cohort was thoughtfully built and analyses & interpretations are sound, solid, and appropriately discussed - I'm sure that it will be well received in the field. Congratulations on a great paper, looking forward to seeing it in print!